# Sensory ASIC3 channel exacerbates psoriatic inflammation via a neurogenic pathway in female mice

Chen Huang[1,2,3,11], Pei-Yi Sun[4,11], Yiming Jiang[2,5,11], Yuandong Liu[6], Zhichao Liu[6], Shao-Ling Han[2], Bao-Shan Wang [2], Yong-Xin Huang[2], An-Ran Ren[2], Jian-Fei Lu[1,2], Qin Jiang[1,2], Ying Li[3], Michael X. Zhu [7], Zhirong Yao [4], Yang Tian [6], Xin Qi [1,2] ✉, Wei-Guang Li [2,8,9,10] ✉ & Tian-Le Xu [1,2,10] ✉

Psoriasis is an immune-mediated skin disease associated with neurogenic inflammation, but the underlying molecular mechanism remains unclear. We demonstrate here that acid-sensing ion channel 3 (ASIC3) exacerbates psoriatic inflammation through a sensory neurogenic pathway. Global or nociceptor-specific *Asic3* knockout (KO) in female mice alleviates imiquimod-induced psoriatic acanthosis and type 17 inflammation to the same extent as nociceptor ablation. However, ASIC3 is dispensable for IL-23-induced psoriatic inflammation that bypasses the need for nociceptors. Mechanistically, ASIC3 activation induces the activity-dependent release of calcitonin gene-related peptide (CGRP) from sensory neurons to promote neurogenic inflammation. Botulinum neurotoxin A and CGRP antagonists prevent sensory neuron-mediated exacerbation of psoriatic inflammation to similar extents as *Asic3* KO. In contrast, replenishing CGRP in the skin of *Asic3* KO mice restores the inflammatory response. These findings establish sensory ASIC3 as a critical constituent in psoriatic inflammation, and a promising target for neurogenic inflammation management.

Psoriasis is a chronic, immune-mediated disorder that affects millions of people worldwide. It is characterized by scaly, erythematous skin lesions that can cause significant discomfort and reduced quality of life[1–3]. The pathological feature of psoriasis is an abnormal and excessive immune response, which involves complex interplays between innate and adaptive immune responses[4,5]. The keratinocytes in the skin epidermis trigger immune responses by producing anti-microbial peptides, S100 proteins, and other effectors upon

[1]Department of Anesthesiology, Songjiang Hospital and Songjiang Research Institute, Shanghai Key Laboratory of Emotions and Affective Disorders, Shanghai Jiao Tong University School of Medicine, Shanghai 201600, China. [2]Department of Anatomy and Physiology, Shanghai Jiao Tong University School of Medicine, Shanghai 200025, China. [3]Basic Medicine Experimental Teaching Center, Shanghai Jiao Tong University School of Medicine, Shanghai 200025, China. [4]Department of Dermatology, Xinhua Hospital, Institute of Dermatology, Shanghai Jiao Tong University School of Medicine, Shanghai 200092, China. [5]Department of Otorhinolaryngology, Renji Hospital, Shanghai Jiao Tong University School of Medicine, Shanghai 200127, China. [6]Shanghai Key Laboratory of Green Chemistry and Chemical Processes, Department of Chemistry, School of Chemistry and Molecular Engineering, East China Normal University, Shanghai 200241, China. [7]Department of Integrative Biology and Pharmacology, McGovern Medical School, The University of Texas Health Science Center at Houston, Houston, TX 77030, USA. [8]Department of Rehabilitation Medicine, Huashan Hospital, Institute for Translational Brain Research, State Key Laboratory of Medical Neurobiology and Ministry of Education Frontiers Center for Brain Science, Fudan University, Shanghai 200032, China. [9]Ministry of Education–Shanghai Key Laboratory for Children's Environmental Health, Xinhua Hospital Affiliated to Shanghai Jiao Tong University School of Medicine, Shanghai 200092, China. [10]Shanghai Research Center for Brain Science and Brain-Inspired Intelligence, Shanghai 201210, China. [11]These authors contributed equally: Chen Huang, Pei-Yi Sun, Yiming Jiang. ✉e-mail: xin597454490@163.com; liwg@fudan.edu.cn; xu-happiness@shsmu.edu.cn

stimulation by various exogenous antigens such as streptococcus and viruses, as well as endogenous self-antigens[6,7]. Additionally, both dendritic cells (DCs) in the skin and keratinocytes secrete various chemokines and cytokines, such as interleukin (IL)−23, to activate γδT lymphocytes[8], which produce effector cytokines such as IL-17 and IL-22. IL-17 mainly recruits neutrophils to mediate inflammatory cell infiltration[9], while IL-22 induces keratinocyte proliferation, resulting in the chronic relapsing nature of psoriasis[10]. Thus, the IL-23/IL-17/IL-22 axis serves as a critical component of a positive feedback loop that promotes psoriasis progression, involving key players such as keratinocytes, DCs, and γδT lymphocytes[11–14].

The involvement of sensory neurons innervating the skin in the pathogenesis of psoriasis is increasingly recognized, with nociceptors emerging as key contributors responsible for pain and itch sensations as well as immune modulation[15,16]. These neurons closely interact with dermal DCs, which play a crucial role in type 17 immune responses in psoriasis[17]. Recently, it has been shown that nociceptors regulate DCs through three distinct mechanisms: release of calcitonin gene-related peptide (CGRP), induction of contact-dependent $Ca^{2+}$ fluxes and membrane depolarization in DCs, and secretion of the chemokine CCL2 to orchestrate local inflammation and induce adaptive responses[18,19]. Intradermal injection of IL-23 bypasses the need for dermal DCs, which are the main source of IL-23, thereby circumventing nociceptor communication with the dermal DCs and directly triggering the IL-23/IL-17/IL-22 pathway to promote psoriatic inflammation[17]. Collectively, the interplay among nociceptor-derived chemokines, neuropeptides, and electrical activity fine-tunes dendritic cell responses, leading to neurogenic inflammation in barrier tissues.

Although significant progresses have been made in understanding the cellular-level interactions between nociceptors and the immune system in psoriatic inflammation, it remains to be elucidated the underlying molecular mechanism that coordinates immune responses in the skin barrier. Identifying the molecular substrates is crucial for gaining valuable insights into the pathogenesis of psoriasis and guiding the development of novel therapies for this irritating disease. As prominent players in the sensory system, transient receptor potential (TRP) ion channels have been implicated in the development of psoriatic inflammation and itch, either in a protective or damaging manner. For instance, TRPV1 has been associated with inflammation susceptibility in an animal model of psoriasis[20], while TRPA1 has shown a protective role in imiquimod (IMQ)-induced psoriasiform dermatitis in mice[21]. Additionally, TRPC4 and TRPV4 have been linked to the pathogenesis of psoriatic inflammation and chronic itch[22,23]. Thus, various sensory ion channels play distinct roles in psoriatic inflammation and require case-by-case investigation.

Acid-sensing ion channels (ASICs) serve as another vanguard of the sensory system that may contribute to psoriatic inflammation. These proton-gated ion channels belong to the degenerin/epithelial $Na^+$ channel (DEG/ENaC) superfamily[24], and they act as extracellular pH sensors. Among them, ASIC3 is prominently expressed in the peripheral nervous system and has been implicated in acidic, inflammatory, and postoperative pain[25,26]. Experimental models have demonstrated upregulation of local ASIC3-immunoreactive nerves in joint inflammation[27]. ASIC3 ablation has been found to reduce scratching behaviors and pathological changes induced by dry skin and concurrent inflammation[28,29]. However, the relationship between ASIC3 and chronic skin inflammation in the context of psoriasis has not been explored. Therefore, this study aims to investigate the contribution of ASIC3 in murine psoriatic skin inflammation. We examined whether ablation of ASIC3 regulates skin phenotype and immune responses, identifying the roles of ASIC3 in the generation and resolution of psoriatic inflammation. Our results suggest that silencing ASIC3 interrupts the pro-inflammatory signaling loop between neural and immune systems, providing evidence for a potential neuroimmune strategy to treat psoriasis and other cutaneous inflammatory diseases.

## Results

### ASIC3 mediates psoriatic skin phenotype and type 17 inflammation

To investigate the role of ASIC3 in psoriatic skin inflammation, we induced psoriatic skin lesions and inflammatory responses by topically applying imiquimod (IMQ) cream to shaved dorsal skin areas of $Asic3^{-/-}$ and $Asic3^{+/+}$ mice on consecutive days (Fig. 1a). We assessed the ensuing inflammatory responses based on changes in spleen weight, skin thickness, proliferation of epidermis, and tissue contents of inflammatory cytokines (Fig. 1a–e). Notably, mice lacking ASIC3 were less susceptible to psoriatic inflammation and displayed more moderate skin lesions than wild types. On day 8, $Asic3^{-/-}$ mice showed less severe splenomegaly than $Asic3^{+/+}$ mice, and they exhibited less pathological changes in the inflamed site (Fig. 1b, c). In particular, in response to the IMQ treatment, $Asic3^{-/-}$ mice displayed significantly lower levels of acanthosis and abnormal epidermal proliferation than $Asic3^{+/+}$ mice as measured by dermal Ki67-positive cell counts (Fig. 1c, d and Supplementary Fig. 1). Moreover, the levels of type 17 cytokines (IL-23, IL-17, and IL-22) in the lesioned skins of $Asic3^{-/-}$ mice were also lower than that in $Asic3^{+/+}$ mice (Fig. 1e). These results indicate that ASIC3 ablation alleviates immunopathological changes in psoriasis by ameliorating keratinocyte hyperproliferation and proinflammatory cytokine production in vivo.

### Selective deletion of ASIC3 in nociceptors suppresses psoriatic skin phenotype and cytokine induction

To investigate the role of ASIC3 in sensory neurons, particularly nociceptors[28,30], in psoriatic inflammation, we crossed a mouse line carrying floxed alleles of $Asic3$ ($Asic3^{flox/flox}$)[31] with $Na_V1.8^{Cre}$ mice[32], enabling selective elimination of $Asic3$ in $Na_V1.8^+$ nociceptors (Supplementary Fig. 2). Following psoriatic modeling, $Na_V1.8^{Cre}::Asic3^{flox/flox}$ mice exhibited lower spleen weight (Fig. 2a) and less epidermal proliferation and infiltrating Ki67+ cells than their $Asic3^{flox/flox}$ siblings (Fig. 2b–e). Furthermore, the progression of Th17 immunity was also attenuated, as indicated by lower expression levels of IL-17, IL-22, and IL-23 (Fig. 2f). These results suggest that the selective loss of ASIC3 in nociceptors effectively alleviates the severity of psoriatic skin lesions and related Th17 inflammation.

### ASIC3 in nociceptors underlies its effects on inflammatory responses in psoriasis

As an alternative strategy to achieve ASIC3 elimination, we generated an adeno-associated virus (AAV) construct that expressed a short hairpin RNA (shRNA) targeting ASIC3, driven by the U6 promoter, along with enhanced green fluorescence protein (EGFP) for visualization (Supplementary Fig. 3a–e). We used the AAV-PHP.S capsid[33] to efficiently and noninvasively deliver either ASIC3-shRNA or negative control vector (NC) to the peripheral nervous system of wild type mice. AAV-ASIC3-shRNA markedly reduced splenomegaly in the psoriasis model (Supplementary Fig. 3f) and mitigated skin inflammation and keratinocyte hyperproliferation, as demonstrated by histopathologic assessment (Supplementary Fig. 3g–j). The ASIC3 knockdown also lessened production of psoriasis-related cytokines, mainly IL-17, IL-22, and IL-23 (Supplementary Fig. 3k). These results further validate the essential role of ASIC3 in the exacerbation of psoriatic inflammation.

We then conducted a cell-type-specific rescue experiment to further confirm the role of ASIC3 in psoriatic inflammation by introducing exogenous ASIC3 in nociceptors together with ASIC3 shRNA (Supplementary Fig. 4a). The AAV vector containing ASIC3 shRNA also carried a double-floxed inverted orientation (DIO) coding sequence for shRNA-resistant FLAG-ASIC3-2A-mCherry (referred to as ASIC3*), which is only expressed in the presence of Cre recombinase. Injecting the AAV into $Na_V1.8^{Cre}$ mice enabled specific re-expression of ASIC3 in $Na_V1.8^+$ nociceptors while maintaining the effectiveness of the shRNA in the Cre-negative wild type mice (Supplementary Fig. 4b, c). Interestingly,

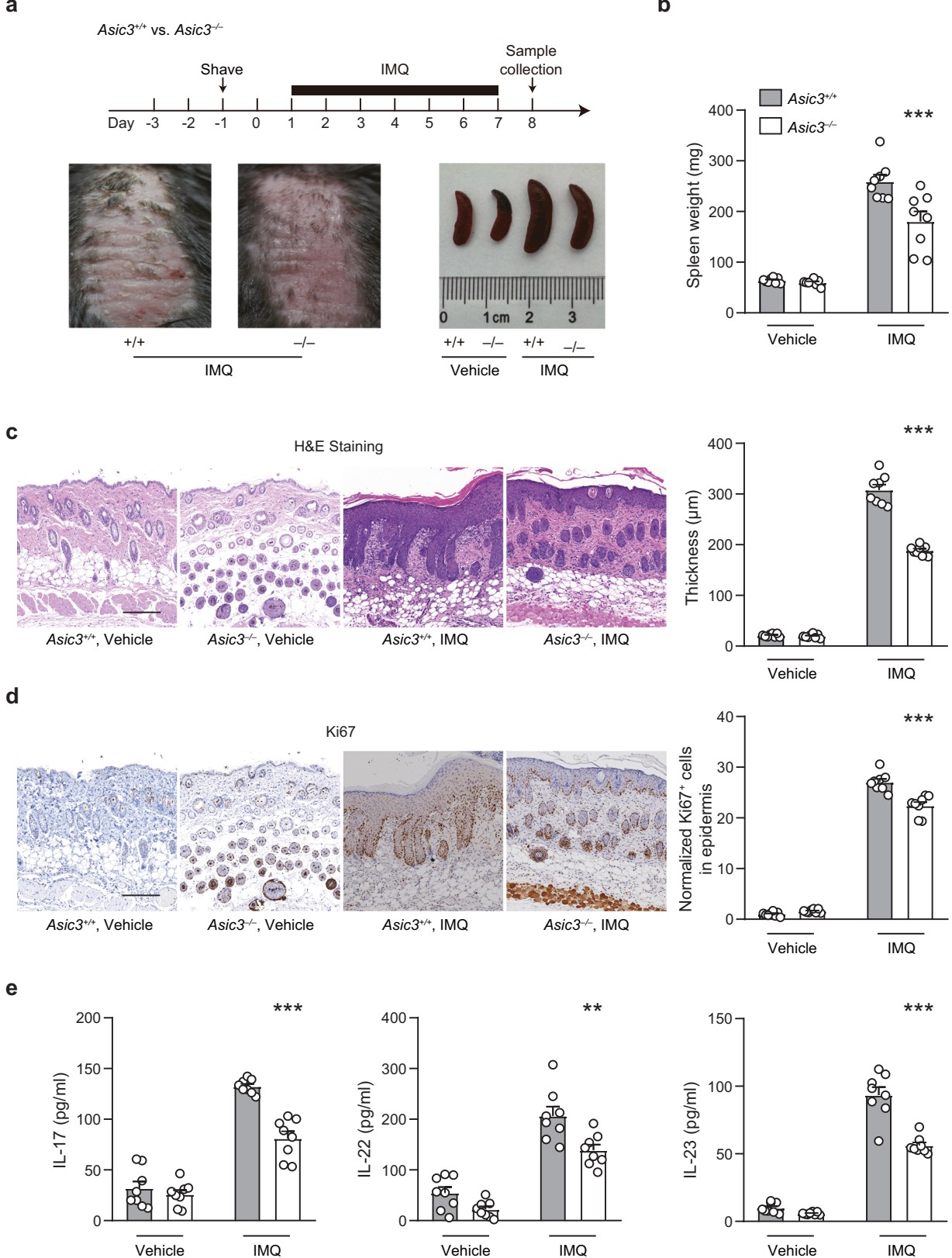

**Fig. 1 | ASIC3 is critical for the development of psoriatic inflammation.**
**a** Experimental schematic indicating imiquimod treatment of $Asic3^{-/-}$ and litter-mate control ($Asic3^{+/+}$) mice and representative photographs of dorsal skin and spleens harvested from the indicated mice. **b** Spleen weight of mice shown in (**a**). $F_{(1,28)} = 11.94$, ***$p = 0.0001$, two-way ANOVA. **c** Representative H&E staining of lesional skin from mice treated with IMQ and quantification of epidermal thickness in $Asic3^{+/+}$ and $Asic3^{-/-}$ mice. Scale bar, 200 μm. IMQ: $F_{(1,28)} = 120.3$, ***$p < 0.001$, $Asic3^{-/-}$ vs. $Asic3^{+/+}$, two-way ANOVA. **d** Representative Ki67 staining of lesional skin from mice treated with IMQ in $Asic3^{+/+}$ and $Asic3^{-/-}$ mice. Scale bar, 200 μm. IMQ: $F_{(1,28)} = 17.42$, ***$p = 4.6063E{-}7 < 0.001$, $Asic3^{-/-}$ vs. $Asic3^{+/+}$, two-way ANOVA. **e** Psoriasis-related cytokine IL-17, IL-22, and IL-23 protein expression in lesional skin of $Asic3^{+/+}$ and $Asic3^{-/-}$ mice. IMQ: IL-17: $F_{(1,28)} = 27.64$, ***$p = 6.2749E{-}7 < 0.001$; IL-22: $F_{(1,28)} = 16.26$, **$p = 0.0012 < 0.01$; IL-23: $F_{(1,28)} = 40.49$, ***$p = 1.6026E{-}8 < 0.001$, $Asic3^{-/-}$ vs. $Asic3^{+/+}$, two-way ANOVA. $n = 8$ mice per group. Summary data are mean ± SEM.

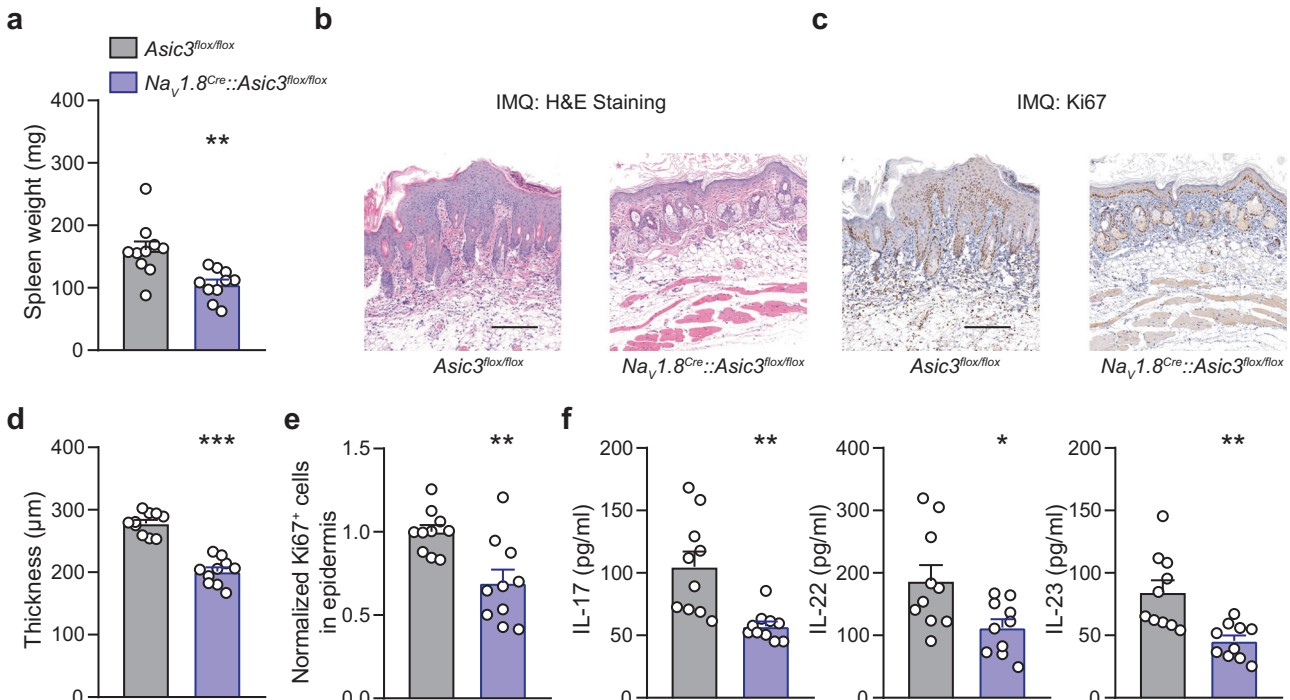

**Fig. 2 | Selective deletion of ASIC3 in nociceptors suppresses skin inflammatory phenotype and cytokine induction in psoriasis. a** Spleen weight of *Asic3^flox/flox* mice and *Na_V1.8^Cre::Asic3^flox/flox* mice. **$p = 0.0027 < 0.01$, two-tailed unpaired Student's *t* test. **b** Representative H&E staining of lesional skin from conditional ASIC3 knockout mice treated with IMQ. Scale bar, 200 μm. **c** Representative Ki67 staining of lesional skin from conditional ASIC3 knockout mice treated with IMQ. Scale bar, 200 μm. **d** Quantification of epidermal thickness in (**b**). ***$p = 5.1451E$ −8 < 0.001, two-tailed unpaired Student's *t* test. **e** Quantification of Ki67 positive cells in (**c**). **$p = 0.0031 < 0.01$, two-tailed unpaired Student's *t* test. **f** Psoriasis-related cytokine IL-17, IL-22, and IL-23 protein expression in lesional skin of *Asic3^flox/flox* and *Na_V1.8^Cre::Asic3^flox/flox* mice. IL-17: **$p = 0.0035 < 0.01$; IL-22: *$p = 0.0175 < 0.05$; IL-23: **$p = 0.0027 < 0.01$, two-tailed unpaired Student's *t* test. $n = 10$ mice per group. Summary data are mean ± SEM.

re-expression of ASIC3 solely in Na_V1.8+ nociceptors aggravated splenomegaly in the psoriasis model (Supplementary Fig. 4d). The conditional rescue of ASIC3 in *Na_V1.8^Cre* mice led to more severe epidermal thickening and Ki67 cell proliferation (Supplementary Fig. 4e–g), as well as higher production of psoriasis-related cytokines, IL-17, IL-22, and IL-23, compared to the ASIC3 knockdown mice (Supplementary Fig. 4h). These results strongly support that ASIC3 in nociceptors is responsible for its role in promoting inflammatory responses in psoriasis.

### *Asic3* KO mice exhibit comparable improvement in psoriatic inflammation as the TRPV1+ nociceptor ablation

We sought to determine whether the role of ASIC3 in psoriatic inflammation was exclusively dependent on its distribution in skin-innervating nociceptors or if additional nociceptor-independent mechanisms were involved. To do so, we generated *Asic3*-myc reporter mice to visualize the distribution of ASIC3 in nociceptors and skin afferents. We observed substantial, but not absolute, co-localization of ASIC3 with TRPV1 in DRG tissues and skin afferents (Supplementary Fig. 5).

To assess the contribution of TRPV1+ nociceptors in ASIC3-dependent skin inflammation in the mouse psoriasis model, we treated *Asic3+/+* and *Asic3−/−* mice with resiniferatoxin (RTX) (Supplementary Fig. 6a). This denervation treatment of TRPV1+ nociceptive terminals significantly reduced splenomegaly of the IMQ-treated *Asic3+/+* mice, indicating that TRPV1+ nociceptive innervation regulates the global inflammatory response in psoriasis. However, in *Asic3−/−* mice, the chemical denervation of TRPV1+ nociceptors failed to further alleviate splenomegaly in the psoriasis model (Supplementary Fig. 6b), arguing for the essential role of ASIC3 in the TRPV1+ nociceptor-mediated regulation. Furthermore, the RTX treatment markedly reduced epidermal hyperplasia and skin thickening in *Asic3+/+* but not *Asic3−/−* mice. Additionally, only in *Asic3+/+* mice, the denervation mitigated Ki67

upregulation in inflamed skin (Supplementary Fig. 6c, d). Thus, the loss of ASIC3 in mice is as effective as nociceptor ablation in ameliorating psoriatic inflammation, underscoring the unequivocal functional significance of ASIC3 in TRPV1+ nociceptors in psoriasis pathogenesis.

### ASIC3 is dispensable for IL-23-induced psoriatic inflammation that bypasses the involvement of nociceptors

Given that nociceptive sensory control of psoriatic inflammation primarily hinges on the intricate interplay between sensory afferents and skin-resident DCs[17], the principal source of IL-23, we next asked whether ASIC3 is still required in IL-23-induced psoriatic inflammation. This experimental approach aims to bypass the nociceptor communication with dermal DCs. Here, we intradermally administered IL-23 in both *Asic3−/−* and *Asic3+/+* mice, with one ear receiving IL-23 and the contralateral ear receiving phosphate-buffered saline (PBS) as a vehicle control (Fig. 3a). The absence of splenomegaly following the local IL-23 injection in both groups indicates that systemic effects were not elicited (Fig. 3b). Notably, both *Asic3−/−* and *Asic3+/+* mice exhibited psoriatic inflammation after IL-23 injection, with histological analysis revealing no statistical differences in inflammatory burden. These included comparable responses in both animal groups in acanthosis, Ki67+ cell infiltration, and elevations of IL-17 and IL-22 levels (Fig. 3c–g). Thus, ASIC3 is not required for nociceptor-independent, IL-23-induced psoriatic inflammation, highlighting the pivotal role of nociceptor-derived ASIC3 in neuroimmune communications to exert the pro-inflammatory effects.

### Peripheral sensory neuron vesicular exocytosis is required for ASIC3 regulation of psoriatic inflammation

Neuropeptides and other active substances released from peripheral sensory nerve terminals typically serve as a means of communicating

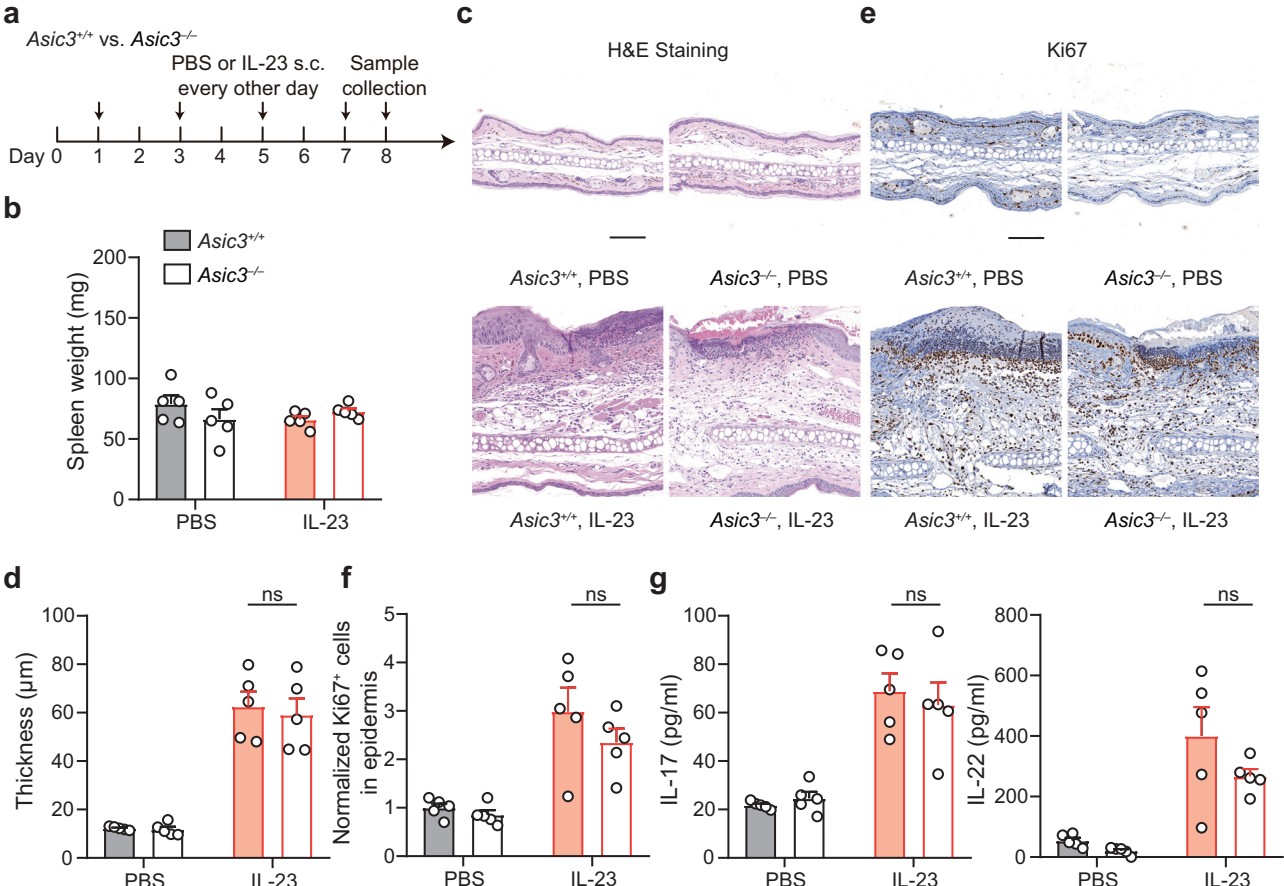

**Fig. 3 | ASIC3 is dispensable for IL-23-induced psoriatic inflammation.**
**a** Schematic protocol of local IL-23 treatment in *Asic3*^+/+ and *Asic3*^−/− mice. **b** Spleen weight of mice in (**a**). $F_{(1,16)} = 0.2450$, $p = 0.6557$, IMQ: *Asic3*^+/+ vs. *Asic3*^−/− mice, two-way ANOVA. **c** Representative sections in lesional skin after IL-23 treatment in *Asic3*^+/+ and *Asic3*^−/− mice. Scale bar, 100 μm. **d** Quantification of epidermal thickness in lesional skin in (**c**). IL-23: $F_{(1,16)} = 0.1841$, $p = 0.8426$, *Asic3*^−/− vs. *Asic3*^+/+, two-

way ANOVA. **e** Representative Ki67 staining in lesional skin. Scale bar, 100 μm.
**f** Ki67^+ cells in *Asic3*^+/+ and *Asic3*^−/− mice after IL-23 modeling in (**e**). IL-23: $F_{(1,16)} = 0.1808$, $p = 0.2616$, *Asic3*^−/− vs. *Asic3*^+/+, two-way ANOVA. **g** Psoriasis-related cytokine IL-17 and IL-22 protein expression in lesional skin. IL-17: $F_{(1,16)} = 0.001796$, $p = 0.7630$; IL-22: $F_{(1,16)} = 0.2899$, $p = 0.1388$, *Asic3*^−/− vs. *Asic3*^+/+, two-way ANOVA. $n = 5$ mice per group. Summary data are mean ± SEM.

sensory neuro-immune interactions[15,18,19,34]. We thus investigated whether ASIC3 regulation of psoriatic inflammation relies on vesicular exocytosis of peripheral neurons. To do so, we injected botulinum neurotoxin A (BoNT/A) locally in *Asic3*^+/+ and *Asic3*^−/− mice (Fig. 4a). BoNT/A is a bacterial toxin that cleaves synaptosomal-associated protein 25 (SNAP-25), a component of the SNARE complex required for vesicle release[35]. We found that BoNT/A prevented the global inflammatory response, as demonstrated by the milder splenomegaly in BoNT/A-treated than the vehicle (PBS)-treated *Asic3*^+/+ mice. By contrast, interdicting neuronal vesicle release by BoNT/A did not alter splenomegaly in *Asic3*^−/− mice (Fig. 4b). Additionally, the BoNT/A treatment alleviated epidermal thickening and keratinocyte hyperproliferation in *Asic3*^+/+ but not *Asic3*^−/− mice (Fig. 4c, d). BoNT/A also attenuated the increase in cytokines, IL-17, IL-22, and IL-23, in *Asic3*^+/+ but not *Asic3*^−/− mice (Fig. 4e). These results indicate that peripheral sensory neuron vesicular exocytosis is involved in ASIC3-dependent regulation of psoriatic inflammation, probably by releasing active substances from the sensory nerve terminals.

## ASIC3 facilitates psoriatic inflammation through activity-dependent CGRP release

CGRP is known to play a crucial role in the development of type 17 inflammation[18,36]; thus, we hypothesized that ASIC3 may drive CGRP release and in turn regulate psoriatic inflammation. To test this hypothesis, we first measured CGRP concentrations in the media of DRG

neuron cultures. Acidic (pH 5.5) stimulation increased CGRP concentration in media collected from *Asic3*^+/+ but not *Asic3*^−/− DRG cultures, which was blocked by the ASIC antagonist amiloride (Fig. 5a). Blocking sensory neuron vesicular exocytosis by BoNT/A also attenuated the acid-induced CGRP release from wild type DRG neurons (Supplementary Fig. 7a). The nonproton agonist of ASIC3, 2-guanidine-4-methylquinazoline (GMQ)[37], also increased CGRP release more robustly in *Asic3*^+/+ than in *Asic3*^−/− DRG cultures (Fig. 5a). The residual effect of GMQ on *Asic3*^−/− cultures may be attributed to its action on A type γ-aminobutyric acid receptors (GABA_ARs)[38] in addition to ASIC3. Consistently, another GABA_AR antagonist, bicuculline also induced CGRP release in an ASIC3-independent manner (Supplementary Fig. 7b). In contrast, high potassium (30 mM KCl) caused comparable increases in CGRP release in both *Asic3*^+/+ and *Asic3*^−/− DRG cultures (Fig. 5b).

We also painted the dorsal skin of *Asic3*^−/− and *Asic3*^+/+ mice with IMQ and measured CGRP levels at the site of inflamed skin using a modified ex vivo skin organ culture 6 h after the last treatment of IMQ[39]. We found that the elevated release of CGRP in IMQ-treated samples was ASIC3-dependent, with skin samples from *Asic3*^−/− mice showing lower CGRP levels than that from the littermate *Asic3*^+/+ controls (Fig. 5c). Additionally, lesions of *Asic3*^−/− mice exhibited lower CGRP^+ and PGP9.5^+ nerve densities than that of *Asic3*^+/+ mice (Fig. 5d, e). Together, the above results suggest that ASIC3 contributes to activity-dependent CGRP release, which may underlie its regulation of psoriatic inflammation.

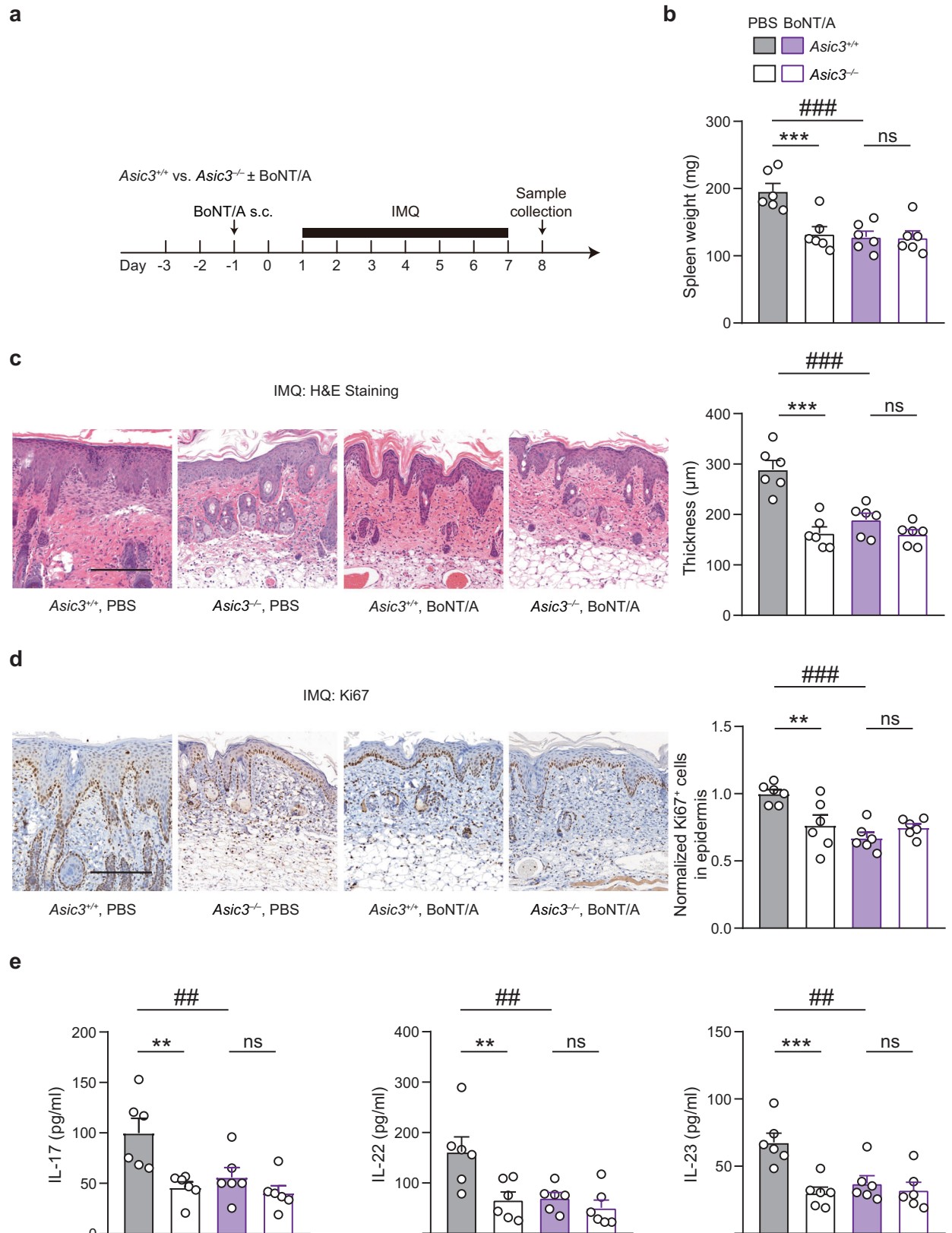

To investigate if CGRP is necessary for ASIC3 regulation of psoriatic immune response, we injected *Asic3*[+/+] mice intradermally with the CGRP antagonist, BIBN4096, prior to the daily IMQ treatment (Fig. 6a). This treatment resulted in inhibition of not only splenomegaly (Fig. 6b), but also epidermal proliferation (Fig. 6c, d) and IL-23, IL-17, and IL-22 protein production (Fig. 6e). Conversely, to verify if CGRP is

sufficient to promote psoriatic inflammation in the absence of other ASIC3-mediated processes, we administered CGRP intradermally prior to the daily IMQ treatment of *Asic3*[−/−] mice for 8 consecutive days. The CGRP treatment led to significant increases in global inflammation (Fig. 6b) and local skin pathological changes in *Asic3*[−/−] mice, with exaggerated epidermal proliferation (Fig. 6c, d) and augmented

**Fig. 4 | Vesicle release of neuropeptides is required for ASIC3-dependent induction of type 17 inflammation in psoriasis. a** Schematic protocol of BoNT/A treatment to block vesicle release of neuropeptides in $Asic3^{+/+}$ and $Asic3^{-/-}$ mice. **b** Spleen weight of mice in (**a**). PBS: $F_{(1,20)} = 9.931$, ***$p = 0.0006 < 0.001$; BoNT/A: $p = 0.9964$, $Asic3^{-/-}$ vs. $Asic3^{+/+}$; $Asic3^{+/+}$: $F_{(1,20)} = 12.89$, ###$p = 0.0003 < 0.001$, BoNT/A vs. PBS, two-way ANOVA. **c** Representative H&E staining (left) and quantification of epidermal thickness (right) of lesional skin from BoNT/A-treated $Asic3^{+/+}$ and $Asic3^{-/-}$ mice. Scale bar, 200 μm. PBS: $F_{(1,20)} = 33.48$, ***$p = 3.4574E{-}6 < 0.001$, $Asic3^{-/-}$ vs. $Asic3^{+/+}$; BoNT/A: $p = 0.2684$, $Asic3^{-/-}$ vs. $Asic3^{+/+}$; $Asic3^{+/+}$: $F_{(1,20)} = 14.40$, ###$p = 7.6417E{-}5 < 0.001$, BoNT/A vs. PBS, two-way ANOVA. **d** Representative Ki67 staining in lesional skin (left) and quantification of Ki67$^+$ cells (right) in BoNT/A-treated $Asic3^{+/+}$ and $Asic3^{-/-}$ mice after psoriasis modeling. Scale bar, 200 μm. PBS: $F_{(1,20)} = 2.744$,

**$p = 0.0048 < 0.01$, $Asic3^{-/-}$ vs. $Asic3^{+/+}$; BoNT/A: $p = 0.4671$, $Asic3^{-/-}$ vs. $Asic3^{+/+}$; $Asic3^{+/+}$: $F_{(1,20)} = 13.09$, ###$p = 0.0002 < 0.001$, BoNT/A vs. PBS, two-way ANOVA. **e** Psoriasis-related cytokine IL-17, IL-22, and IL-23 protein expression in lesional skin after BoNT/A treatment. IL-17: $F_{(1,20)} = 12.63$, PBS: **$p = 0.0018 < 0.01$, $Asic3^{-/-}$ vs. $Asic3^{+/+}$; BoNT/A: $p = 0.4720$, $Asic3^{-/-}$ vs. $Asic3^{+/+}$; $Asic3^{+/+}$: $F_{(1,20)} = 6.342$, ##$p = 0.0097 < 0.01$, BoNT/A vs. PBS, two-way ANOVA. IL-22: PBS: $F_{(1,20)} = 8.982$, **$p = 0.0044 < 0.01$, $Asic3^{-/-}$ vs. $Asic3^{+/+}$; BoNT/A: $p = 0.7271$, $Asic3^{-/-}$ vs. $Asic3^{+/+}$; $Asic3^{+/+}$: $F_{(1,20)} = 7.720$, ##$p = 0.0062 < 0.01$, BoNT/A vs. PBS, two-way ANOVA. IL-23: PBS: $F_{(1,20)} = 13.67$, ***$p = 0.0003 < 0.001$, $Asic3^{-/-}$ vs. $Asic3^{+/+}$; BoNT/A: $p = 0.8025$, $Asic3^{-/-}$ vs. $Asic3^{+/+}$; $Asic3^{+/+}$: $F_{(1,20)} = 6.115$, ##$p = 0.0024 < 0.01$, BoNT/A vs. PBS, two-way ANOVA. $n = 6$ mice per group. Summary data are mean ±SEM.

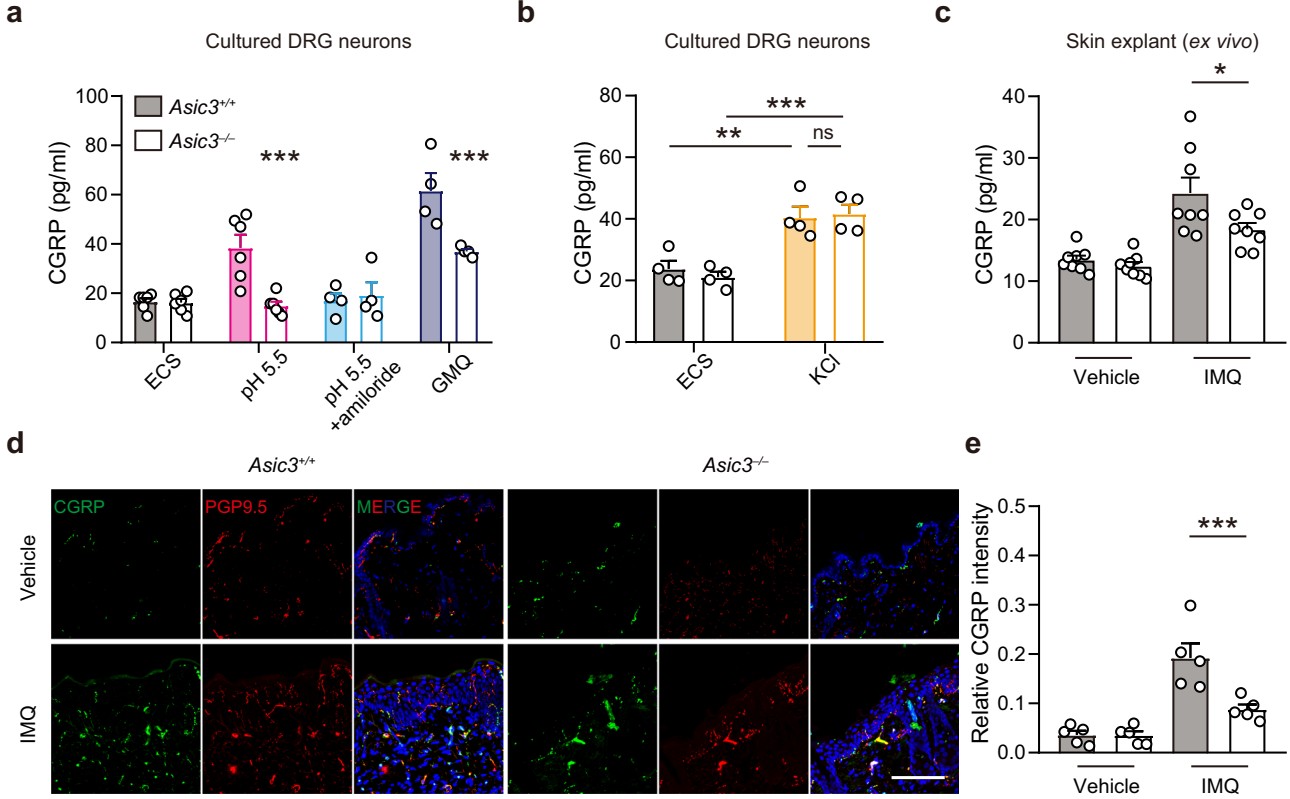

**Fig. 5 | ASIC3 drives CGRP release in psoriatic inflammation. a** DRG neurons from $Asic3^{+/+}$ and $Asic3^{-/-}$ mice were stimulated with acid (pH 5.5), acid plus amiloride (100 μM amiloride in pH 5.5) or GMQ (1 mM GMQ) for 30 min and CGRP was measured in the extracellular solution (ECS). $n = 6$ for ECS and pH 5.5 groups and $n = 4$ for pH 5.5 + AMI and GMQ groups. $F_{(1,32)} = 19.64$, pH 5.5: ***$p < 0.001$; pH 5.5 + AMI: $p = 0.9925$; GMQ: ***$p = 0.0006 < 0.001$, $Asic3^{-/-}$ vs. $Asic3^{+/+}$, two-way ANOVA. **b** DRG neurons from $Asic3^{+/+}$ and $Asic3^{-/-}$ mice were stimulated with high potassium (30 mM KCl) for 30 min and CGRP was measured in neuronal supernatant. $n = 4$ for ECS and KCl groups. $F_{(1,12)} = 0.06706$, KCl: $p = 0.9462$, $Asic3^{-/-}$ vs.

$Asic3^{+/+}$; $F_{(1,12)} = 44.40$, $Asic3^{+/+}$: **$p = 0.0023 < 0.01$, ECS vs. KCl, $Asic3^{-/-}$: ***$p = 0.0004 < 0.001$, ECS vs. KCl; two-way ANOVA. **c** CGRP release ex vivo from skin punch biopsies of $Asic3^{+/+}$ and $Asic3^{-/-}$ mice treated with IMQ. $n = 8$ mice per group. IMQ: $F_{(1,28)} = 5.869$, *$p = 0.0131 < 0.05$, $Asic3^{-/-}$ vs. $Asic3^{+/+}$, two-way ANOVA. **d** Representative images of skin tissues isolated from $Asic3^{+/+}$ and $Asic3^{-/-}$ mice treated with vehicle or IMQ and stained with anti-CGRP (green), anti-PGP9.5 (red) and DAPI (blue). Scale bar, 100 μm. **e** Relative CGRP intensity normalized by DAPI in (**d**). IMQ: $F_{(1,16)} = 9.945$, ***$p = 0.0008 < 0.001$, $Asic3^{-/-}$ vs. $Asic3^{+/+}$, two-way ANOVA. $n = 5$ for each group. Summary data are mean ± SEM.

production of type 17 cytokines, just like the wild type mice (Fig. 6e). Based on these results, we conclude that ASIC3 exacerbates psoriatic inflammation by regulating activity-dependent CGRP release.

**ASIC3-dependent CGRP signaling activates DCs and promotes keratinocyte proliferation to drive psoriatic inflammation**

Given that cutaneous immune cells express various neuropeptide receptors, including CGRP receptors that facilitate the transduction of neuronal signals to immune responses[36], we examined the potential connection between CGRP and dendritic cell activation. We established a co-culture system including DRG neurons and bone marrow-derived DCs (BMDCs). Following the co-culture with

DRG neurons, BMDCs were collected and evaluated by flow cytometry. In co-cultures of $Asic3^{+/+}$ DRG neurons and BMDCs, acid treatment (pH 5.5) led to an upregulation of IL-23$^+$ BMDCs (also CD11c$^+$ and major histocompatibility complex II$^+$), which was reversed with the inhibition of CGRP receptors by BIBN4096. Notably, this effect was absent in co-cultures of $Asic3^{-/-}$ DRG neurons and BMDCs (Supplementary Fig. 8). As a control, IL-23 production induced by the combined inflammatory stimuli, lipopolysaccharide (LPS) and the toll-like receptor 7/8 (TLR7/8) agonist R848, was comparable from BMDCs co-cultured with either $Asic3^{+/+}$ or $Asic3^{-/-}$ DRG neurons, which was also effectively inhibited by BIBN4096 (Supplementary Fig. 8).

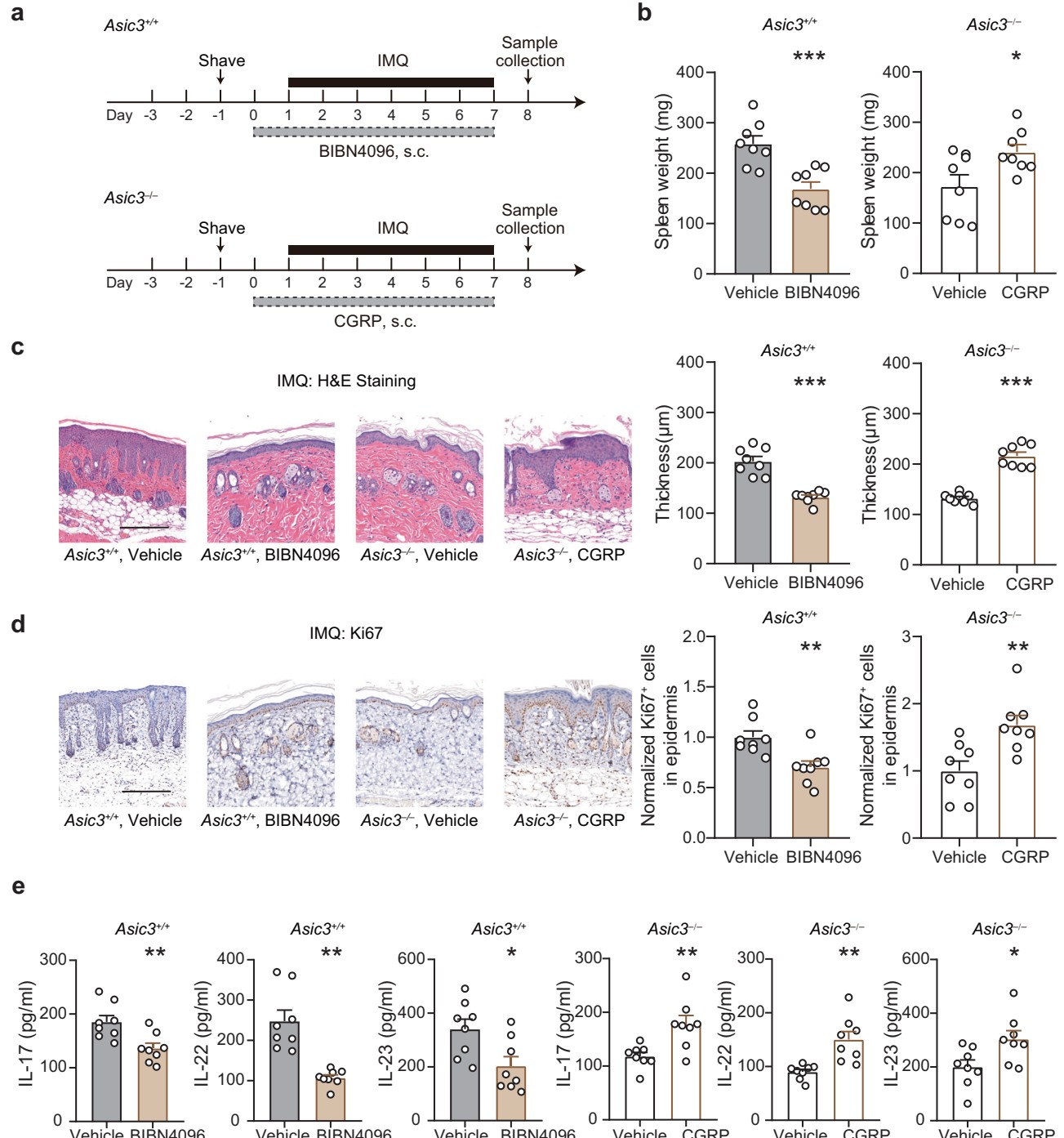

**Fig. 6 | ASIC3 drives psoriatic inflammation through CGRP release. a** Schematic protocol of CGRP receptor antagonism in *Asic3*[+/+] mice and local CGRP supplement in *Asic3*[−/−] mice. **b** Spleen weight of mice in (**a**). *Asic3*[+/+]: ***$p$ = 0.0008 < 0.001, vehicle vs. BIBN4096, two-tailed unpaired Student's $t$ test. *Asic3*[−/−]: *$p$ = 0.0269 < 0.05, vehicle vs. CGRP, two-tailed unpaired Student's $t$ test. **c** Representative H&E staining (left) and quantification of epidermal thickness (middle and right) of lesional skin after CGRP receptor antagonism in *Asic3*[+/+] and local CGRP supplement in *Asic3*[−/−] mice. Scale bar, 200 μm. *Asic3*[+/+]: ***$p$ = 5.6092E −5 < 0.001, vehicle vs. BIBN4096, two-tailed unpaired Student's $t$ test. *Asic3*[−/−]: ***$p$ = 2.5174E−6 < 0.001, vehicle vs. CGRP, two-tailed unpaired Student's $t$ test. **d** Representative Ki67 staining (left) and quantification of Ki67[+] cells (middle and

right) in lesional skin after CGRP receptor antagonism in *Asic3*[+/+] and local CGRP supplement in *Asic3*[−/−] mice. Scale bar, 200 μm. *Asic3*[+/+]: **$p$ = 0.0043 < 0.01, vehicle vs. BIBN4096, two-tailed unpaired Student's $t$ test. *Asic3*[−/−]: **$p$ = 0.0049 < 0.01, vehicle vs. CGRP, two-tailed unpaired Student's $t$ test. **e** Psoriasis-related cytokine IL-17, IL-22, and IL-23 protein expression in lesional skin after CGRP receptor antagonism in *Asic3*[+/+] and local CGRP supplement in *Asic3*[−/−] mice. *Asic3*[+/+]: IL-17: **$p$ = 0.0060 < 0.01; IL-22: **$p$ = 0.0011 < 0.01; IL-23: *$p$ = 0.0167 < 0.05, vehicle vs. BIBN4096, two-tailed unpaired Student's $t$ test. *Asic3*[−/−]: IL-17: **$p$ = 0.0051 < 0.01; IL-22: **$p$ = 0.0043 < 0.01; IL-23: *$p$ = 0.0232 < 0.05, vehicle vs. CGRP, two-tailed unpaired Student's $t$ test. $n$ = 8 mice per group. Summary data are mean ± SEM.

Considering the concomitant feature of epidermal proliferation with psoriasis, we also examined the direct effect of CGRP on keratinocyte proliferation. Indeed, CGRP induced a time-dependent increase in the proliferation of human keratinocytes, HaCaT cells (Supplementary Fig. 9). Therefore, ASIC3-dependent CGRP signaling leads to DC-derived immune responses and keratinocyte proliferation, providing novel insights into the neuroimmune mechanism underlying this specific pathological scenario.

## Activation of ASIC3 under psoriatic conditions by acidosis and lipid mediators

Finally, to learn how ASIC3 is activated in the context of psoriatic inflammation, we measured the pH of psoriatic lesions using an electrochemical biosensor[40]. In stark contrast to the vehicle-treated skin, which maintained relatively constant pH levels from the 1st to the 7th day, psoriatic lesions exhibited a gradual decrease in pH following the daily IMQ treatment (Supplementary Fig. 10). The acidic shift observed in the area affected by psoriatic inflammation strongly suggests the presence of acidic metabolites that may function as a trigger for ASIC3 activation, thereby establishing the neuroimmune interaction.

Moreover, our lipidomic analysis of IMQ-induced psoriatic lesions revealed a distinct lipid profile compared to skin samples from the vehicle controls (Supplementary Fig. 11). Specifically, lysophosphatidylcholine (LPC) was markedly up-regulated at subclass levels, with LPC 14:0 ($p < 0.001$), 16:1 ($p < 0.001$), 18:2 ($p = 0.0277 < 0.05$), 18:3 ($p = 0.0368 < 0.05$) and 20:3 ($p = 0.0285 < 0.05$) emerging as the significantly elevated molecular species based on multiple comparisons with Benjamini-Hochberg adjustment using the raw lipidomic data after probabilistic quotient normalization and autoscaling (Supplementary Fig. 12a). That not all molecular species of LPC were significantly changed by the IMQ treatment suggests that the various LPC molecular species are not evenly or indiscriminately produced in the psoriatic skin.

To learn how LPC affects ASIC3 function, we used LPC14:0, which showed the most increase in percentage (78.3%) in the psoriatic skin (Fig. 7a). Notably, LPC14:0 ($10\,\mu M$) not only sensitized acid-evoked currents (Supplementary Fig. 12b, c) but at a higher concentration ($30\,\mu M$) also evoked currents at pH 7.4 in Chinese hamster ovary (CHO) cells that heterologously expressed ASIC3 (Fig. 7b, c). In addition, LPC14:0 ($30\,\mu M$) not only increased CGRP release at pH 7.4 but also enhanced CGRP release induced by pH 6.5 in $Asic3^{+/+}$ DRG cultures, effects not seen in $Asic3^{-/-}$ DRG cultures (Fig. 7d). Given their relative abundance and reported function on ASIC3[41,42], we also tested whether LPC16:0, LPC18:0 and LPC18:1 could induce CGRP release in $Asic3^{+/+}$ and $Asic3^{-/-}$ DRG cultures. The results showed that LPC18:0 and LPC18:1, but not LPC16:0, induced CGRP release in $Asic3^{+/+}$ DRG cultures at neutral pH without enhancing the acid-induced CGRP release (Supplementary Fig. 12d), further supporting the subclass-dependence of the LPC molecular species on ASIC3-mediated effects. To explore the in vivo effect of LPC, subcutaneous injection of LPC (14:0) was made, which resulted in epidermal proliferation and enhanced production of Th17 cytokines in $Asic3^{+/+}$ but not $Asic3^{-/-}$ mice (Fig. 7e–i). Collectively, our findings suggest that LPC can potentially serve as triggers in ASIC3-dependent exacerbation of psoriatic inflammation. This occurs through orchestration of activity-dependent CGRP release, leading to DC-derived immune responses and keratinocyte proliferation, providing novel insights into the neuroimmune mechanism in this pathological scenario (Fig. 8).

## Discussion

The skin as a physical barrier and sensory interface is constantly exposed to external stimuli that can trigger inflammatory responses[43], with nociceptive neurons and immune cells working in concert to regulate these responses in various skin disorders, including psoriasis. Recent studies have documented the roles of neuroimmune interaction at barrier surface in tissue inflammation[43], and skin-innervating nociceptors have been reported to drive skin inflammation in mouse models of skin disorders, including psoriasis[17], contact dermatitis[44], and pathogen infections[36,39,45,46]. While most previous studies pinpoint to the central role of TRPV1+ nociceptors in regulating cutaneous immune responses, the molecular underpinnings of such neuroimmune interaction remain unclear. Our current study shows that ASIC3, an ion channel that was previously neglected in skin inflammation but significantly implicated in itch and nociception[25,28,29,47,48], contributes to psoriatic skin phenotype and the accompanying type 17 inflammation. Our results demonstrated that global or nociceptor-specific knockout of $Asic3$ in mice significantly reduced psoriatic phenotype and type 17 inflammation induced by IMQ to the same degree as nociceptor ablation. However, ASIC3 is not required for IL-23-induced psoriatic inflammation, which bypasses the involvement of nociceptors. We further observed that ASIC3 activation led to activity-dependent release of CGRP from sensory neurons, and that botulinum neurotoxin A and CGRP antagonist prevented sensory neuron-mediated exacerbation of psoriatic inflammation, comparable to $Asic3$ KO. Conversely, replenishing CGRP in $Asic3$ KO skin restored the inflammatory response. Overall, our findings establish sensory ASIC3 as a critical player of psoriatic inflammation and a promising target for managing chronic skin diseases, akin to cellular blockade of neuroimmune communication.

In the intricate interplay between cutaneous nociceptors and dermal DCs, the role of sensory ASIC3 channels in psoriatic inflammation emerges as plausible, although the precise mechanism remains incompletely understood[17,36]. Particularly, TRPV1+ nociceptive neurons, known for transmitting electrical signals from sites of infection or inflammation to neighboring skin through antidromic neuron activation, contribute to an "anticipatory immunity"[18]. Within this context, ASIC3, as a Na+-selective ion channel in nociceptors, may participate in the generation, propagation, or amplification of electrical signals, potentially exacerbating psoriatic inflammation. Our construction of ASIC3 reporter mice yielded compelling evidence supporting the colocalization of ASIC3 with TRPV1 in skin afferents, implicating their interactions in the nociceptive sensory control of psoriatic inflammation. Furthermore, as ASIC3 protein serves as a component of a mechanoreceptor or nociceptor complex[49], it becomes integral for engaging in anticipatory immunity in psoriatic inflammation.

The intricate interactions between nociceptors and dermal DCs in barrier tissues, which are crucial in modulating immune responses, involve three molecularly distinct mechanisms: nociceptor-mediated CGRP release, induction of contact-dependent $Ca^{2+}$ fluxes and membrane depolarization in DCs, and secretion of the chemokine CCL2 to orchestrate local inflammation and induce adaptive responses[19]. Our study on psoriatic inflammation provides strong evidence that ASIC3 is a pivotal player in activity-dependent CGRP release, both in vivo and in vitro. This finding aligns with a previous study[50] demonstrating that ASIC3, distinct form TRPV1, can trigger acidification-evoked, $Ca^{2+}$-independent CGRP secretion from cultured trigeminal neurons. These results suggest that ASIC3 mediates a unique form of CGRP release, distinct from the $Ca^{2+}$ and SNAP25-dependent pathway, likely contributing significantly to the nociceptor-DC interaction under conditions of psoriatic inflammation.

However, our findings also reveal that blockade of SNAP25-dependent peripheral sensory neuron vesicular exocytosis alleviates psoriatic inflammation to a comparable degree as $Asic3$ KO. Strikingly, it fails to further improve the skin phenotype and type 17 inflammation in the $Asic3$ KO mouse model. This implies that, irrespective of the CGRP release mechanism (either SNAP25-dependent and -independent), the quantity of CGRP may be the major determinant for the pathogenesis of psoriasis. In this specific scenario, ASIC3-dependent CGRP signaling propels DC-derived immune activation, leading to the release of IL-23. Moreover, CGRP also directly promotes keratinocyte

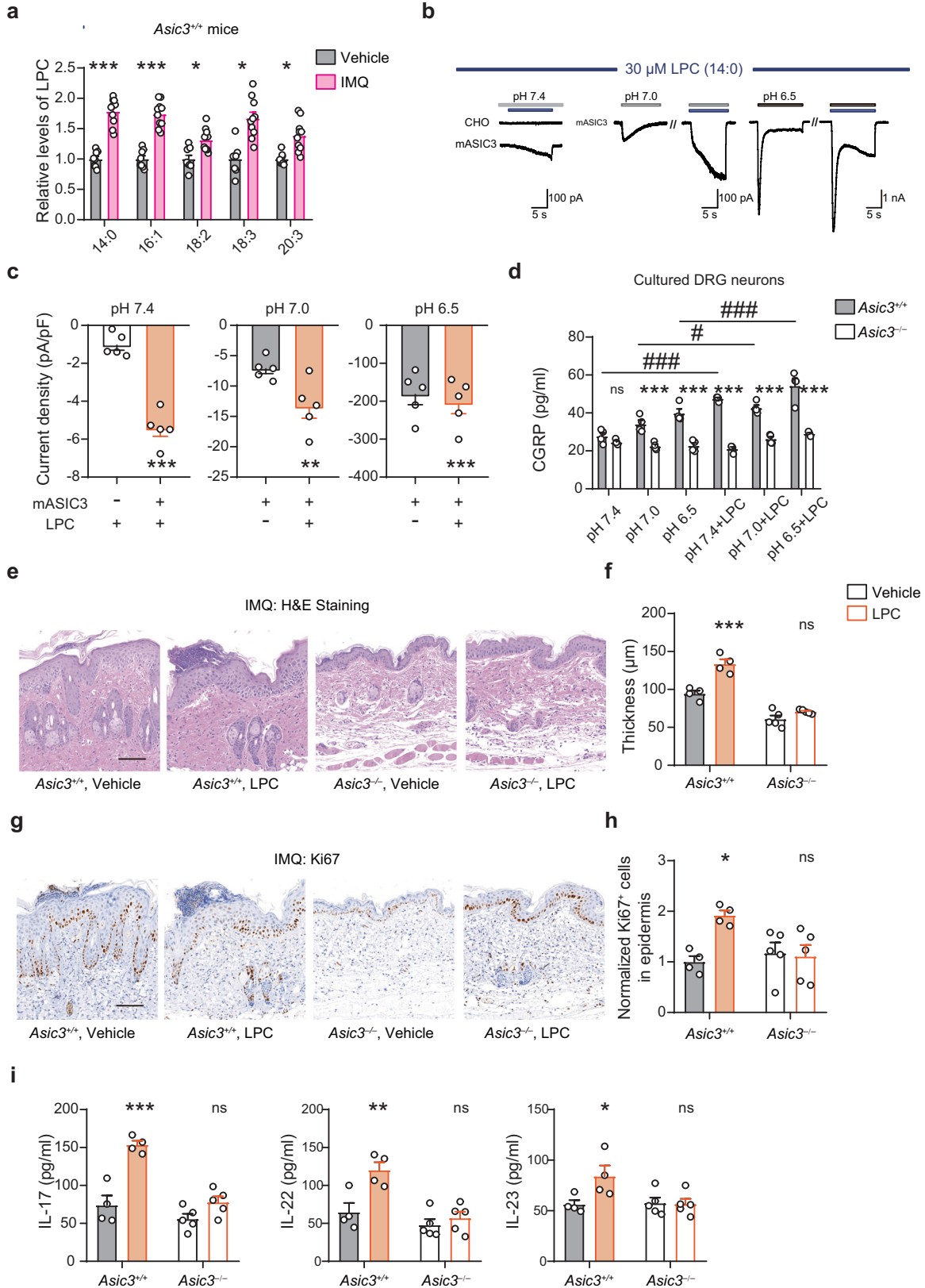

proliferation. Collectively, these processes contribute to the intricate mechanisms underlying psoriatic inflammation.

Furthermore, our observations reveal a consistent and substantial increase in the density of CGRP[+] nerve fibers in psoriasis-modeled skin compared to healthy controls, a phenomenon significantly reversed in *Asic3*[−/−] mice. This not only suggests that heightened neurite

outgrowth results in elevated neuropeptide levels, amplifying the neuroimmune loop in psoriasis conditions, but also implies a potentially increased direct contact between nociceptors and DCs. The genetic ablation of ASIC3 not only reversed the ectopic sprouting of nerve fibers but also remodeled these fibers to a resting state, further supporting the involvement of ASIC3 in such direct interactions.

**Fig. 7 | LPC promotes ASIC3 activation in psoriatic inflammation. a** Relative levels of LPC molecular species increased in skin samples of *Asic3*[+/+] mice after the IMQ treatment. For each species, the data were normalized to that of the vehicle control. Statistical analysis used raw data as described in Supplementary Fig. 12a. $F_{(1,304)}$ = 39.71, LPC (14:0): ***$p$ = 1.3268E−7 < 0.001; LPC (16:1): ***$p$ = 3.1771E −7 < 0.001; LPC (18:2): *$p$ = 0.0277 < 0.05; LPC (18:3): *$p$ = 0.0368 < 0.05; LPC (20:3): *$p$ = 0.0285 < 0.05); multiple comparisons with Benjamini-Hochberg adjustment. **b** Whole-cell currents. Bath solution was changed to pH 7.4, pH 7.0 and pH 6.5 containing LPC14:0 (30 μM) or not as indicated in the horizontal bars. **c** Peak current density without leak subtraction in response to exposure to bath solutions of different pH before and after applications of 30 μM LPC14:0 in CHO cells transfected with *mAsic3*. pH 7.4: ***$p$ = 4.1232E−5 < 0.001, $n$ = 5; pH 7.0: **$p$ = 0.0097 < 0.01, $n$ = 5; pH 6.5: ***$p$ = 0.0009 < 0.001, $n$ = 5; two-side paired *t*-tests. **d** CGRP levels in supernatant of DRG neurons. $n$ = 4 for each group. $F_{(1,36)}$ = 260.9, pH 7.4: $p$ = 0.7819; pH 7.0: ***$p$ = 0.0003 < 0.001; pH 6.5: ***$p$ = 3.7503E−7 < 0.001; pH 7.4 + LPC: ***$p$ = 1.0722E−11 < 0.001; pH 7.0 + LPC:

***$p$ = 9.5219E−7 < 0.001; pH 6.5 + LPC: ***$p$ = 3.8016E−11 < 0.001; *Asic3*[−/−] vs. *Asic3*[+/+]; *Asic3*[+/+]: $F_{(5,36)}$ = 18.94, ###$p$ = 3.7748E−8 < 0.001, pH 7.4 vs. pH 7.4 + LPC; #$p$ = 0.0268 < 0.05, pH 7.0 vs. pH 7.0 + LPC; ###$p$ = 2.2664E−5 < 0.001, pH 6.5 vs. pH 6.5 + LPC; two-way ANOVA. Representative sections (**e**) and quantification of epidermal thickness (**f**) in lesional skins after LPC treatment. Scale bar, 100 μm. $F_{(1,14)}$ = 33.65, *Asic3*[+/+]: ***$p$ = 4.4806E−5 < 0.001, vehicle vs. LPC; *Asic3*[−/−]: $p$ = 0.1943, vehicle vs. LPC; two-way ANOVA. Representative Ki67 staining (**g**) and number of Ki67[+] cells (**h**) in lesional skins. Scale bar, 100 μm. $F_{(1,14)}$ = 5.187, *Asic3*[+/+]: *$p$ = 0.0109 < 0.05, vehicle vs. LPC; *Asic3*[−/−]: $p$ = 0.9617, vehicle vs. LPC; two-way ANOVA. **i** Psoriasis-related cytokine. IL-17: $F_{(1,14)}$ = 36.89, *Asic3*[+/+]: ***$p$ = 3.4388E −5 < 0.001; *Asic3*[−/−]: $p$ = 0.1307, vehicle vs. LPC; two-way ANOVA. IL-22: $F_{(1,14)}$ = 11.59, *Asic3*[+/+]: **$p$ = 0.0031 < 0.01; *Asic3*[−/−]: $p$ = 0.7313, vehicle vs. LPC; two-way ANOVA. IL-23: $F_{(1,14)}$ = 4.245, *Asic3*[+/+]: *$p$ = 0.0253 < 0.05; *Asic3*[−/−]: $p$ = 0.9937, vehicle vs. LPC; two-way ANOVA. $n$ = 4 and 5 for the *Asic3*[+/+] and *Asic3*[−/−] groups. Summary data are mean ± SEM.

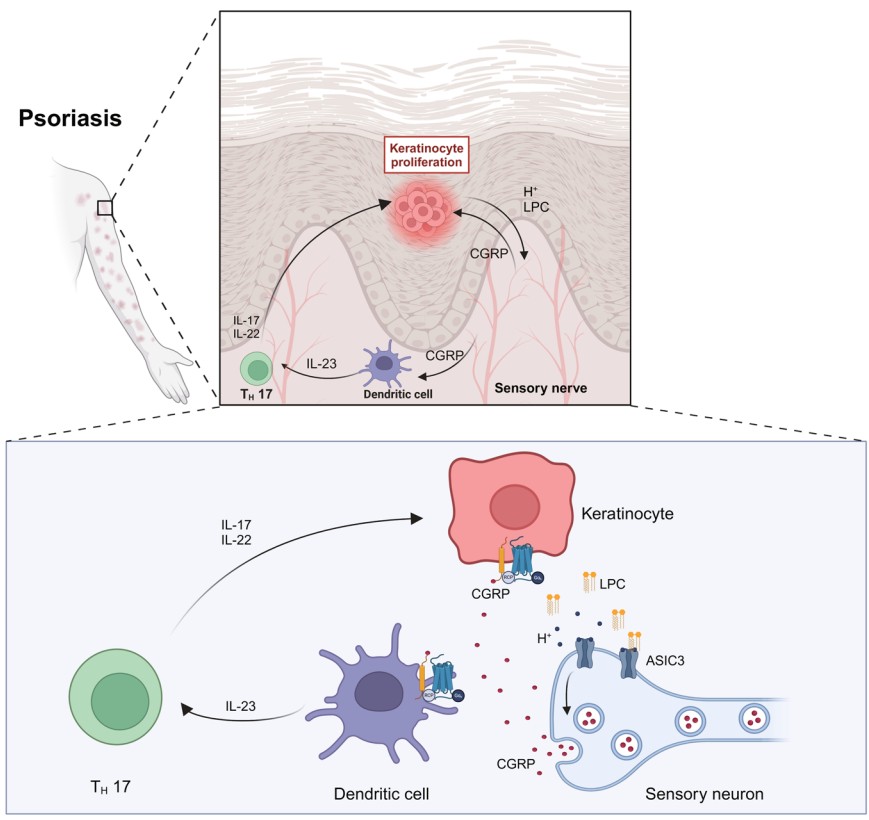

**Fig. 8 | Working model of nociceptor ASIC3-dependent regulation of psoriatic inflammation.** ASIC3-expressing nociceptors innervating the skin contribute to the psoriatic skin phenotype and type 17 immunity through the neuropeptide CGRP. CGRP can both promote IL-23 production from DCs and induce keratinocyte proliferation. IL-23 acts downstream of ASIC3-dependent CGRP release and exacerbates inflammatory response. Acid and LPC serve as potential triggers that activate ASIC3 in psoriatic inflammation. ASIC3 orchestrates the neuroimmune crosstalk in the progression of psoriatic inflammation. The Figure Created with BioRender.com released under a Creative Commons Attribution-NonCommercial-NoDerivs 4.0 International license.

Beyond its expression in sensory neurons, ASIC3 has been reported to be functionally expressed in DCs[51], where it promotes dendritic cell maturation. Future studies should explore the role of ASIC3 in DCs in the context of psoriatic inflammation.

Concerning the potential mechanism underlying ASIC3 activation in psoriatic inflammation, we identified significant tissue acidosis in IMQ-induced psoriatic lesions compared to vehicle controls. This finding aligns with existing literature showing that psoriatic patients have an overall lower skin pH than healthy subjects[52]. In addition to acidosis, our characterization of the distinct lipid metabolomics profile in the IMQ-induced psoriasis model revealed upregulation of several molecular species of LPC. This observation is also consistent with findings in humans, where LPC levels were significantly elevated in

lesioned psoriatic skin compared with uninvolved and normal skin[53,54]. Importantly, our results, along with previous reports, indicate that LPC can evoke and enhance ASIC3 currents in the absence and presence of acidosis, respectively[41,42,55]. Strikingly, in vitro application of LPC14:0 enhanced CGRP release in the absence and presence of acidosis from *Asic3*[+/+] but not *Asic3*[−/−] DRG neurons, and in vivo local injection of LPC intensified psoriatic inflammatory and pathological phenotype. These findings support that at least certain LPC species, in conjunction with acidosis, significantly contribute to the induction and maintenance of inflammatory and immunological processes through ASIC3 in lesioned psoriatic skin. It is noteworthy that ASIC3 can be upregulated, potentiated, or even activated by various inflammatory mediators, such as serotonin (5-HT), bradykinin (BK), and nerve growth factor[56–58], which

are all implicated in the pathogenesis and activity of psoriasis[59–61]. ATP, a co-agonist[62] of ASIC3, has also been identified to contribute to the development of psoriasiform dermatitis[63]. The mechanosensitive nature[31,49,64–66] of ASIC3 may play an additional role in tuning the dynamic interactions between nociceptors and immune cells. Overall, the multitude of immune and inflammatory pathways activating ASIC3 in psoriatic inflammation underscores its significance as a therapeutic target for chronic inflammatory diseases.

In conclusion, our study unveils novel insights into the mechanism governing neuroimmune interactions in psoriasis, emphasizing the therapeutic potential of targeting ASIC3 in chronic inflammatory skin diseases. The scope of ASIC3-mediated inflammatory conditions, spanning pruritus[28], atopic dermatitis[67], and pain associated with arthritis[27,42,68,69], is broadened by our findings. This suggests that modulating ASIC3 holds promise for therapeutic applications beyond pain and itch sensations in various inflammatory contexts. These results underscore the importance of molecular understanding of the intricate interplay between sensory neurons and immune cells in tissue inflammation, which paves the way for the development of innovative therapeutic strategies for psoriasis and other inflammatory disorders.

# Methods

## Mice
C57BL/6 mice were obtained from Shanghai Slac Laboratory Animal Company, China. To minimize experimental variability, age-matched littermate pairs resulting from heterozygous crossings were used for all experiments. $Asic3^{-/-}$ mice were prepared as previously described[28]. To achieve nociceptor-specific deletion of $Asic3$, $Na_v1.8^{Cre}$ mice were bred with $Asic3^{flox/flox}$ mice. The $Na_v1.8^{Cre}$ mice, generated by Professor Rohini Kuner[32], were generously provided by Professor Xu Zhang at Shanghai Advanced Research Institute, Chinese Academy of Sciences, Shanghai, China, and the $Asic3^{flox/flox}$ mice[31] were provided by Professor Chih-Cheng Chen at Institute of Biomedical Sciences, Academia Sinica, Taipei, Taiwan. We generated $Asic3-myc$ mice by CRISPR/Cas9-mediated genome engineering. The 3×Myc tag was inserted downstream of ATG start codon in exon 1 of the mouse $Asic3$ gene. All experiments were performed using 6-week-old female mice. All mice were bred in specific pathogen-free laboratory animal facilities under standard conditions with temperatures of 21–23 °C, 40–60% humidity and 12 h light/dark cycles, with rodent chow and water ad libitum. Animal care and experimental protocols were approved by the Animal Ethics Committee of Shanghai Jiao Tong University School of Medicine, Shanghai, China. All animal experimental designs conform to the ARRIVE guideline 2.0.

## Animal models of psoriasis and treatment
The dorsal skin of female mice, of indicated genotypes or pharmacological intervention, was shaved and allowed to recover for 1 day. Mice that showed hair regrowth were excluded. The remaining mice were treated with a daily topical dose of 62.5 mg of 5% IMQ cream (Sichuan Med-shine Pharmaceutical, Catalog no. H20030128) or 62.5 mg Vaseline cream for the control group by cotton swab in a constrained area (3 cm by 2 cm) on the shaved back for 7 consecutive days. For the IL-23-induced mouse model of psoriasis, recombinant mouse IL-23 (R&D Systems, Catalog no. 1187-ML-010) was injected intradermally into one ear as a 50 μg ml⁻¹ solution in 20 μl PBS (1 μg per ear). Contralateral ears were injected with 20 μl PBS alone. Injections were given every other day for a total of 8 days (days 1, 3, 5, and 7). For local pretreatment, different groups of mice received a subcutaneous 30 U kg⁻¹ BoNT/A injection or vehicle in the dorsal skin at the anticipated site of psoriatic inflammation. To treat $Asic3^{+/+}$ mice, a daily subcutaneous administration of 10 μg BIBN4096 (Tocris, Catalog no. 4561) was delivered to the anticipated inflamed dorsal skin and the same volume of PBS was injected as a control. To examine the effects of CGRP on psoriatic

inflammation, a daily subcutaneous injection of 0.5 μg CGRP (Tocris, Catalog no. 1161) was made to the anticipated inflamed site in dorsal skin of $Asic3^{-/-}$ mice. The pharmacological experiments with CGRP and antagonists in vivo were performed on day 0 and then 2 h prior to the daily IMQ treatment and PBS as a vehicle was given in the same manner. On the eighth day, the dorsal skin or ears were harvested and analyzed by histopathology and cytokine quantification.

## Denervation
For chemical TRPV1⁺ nociceptor denervation, 100 μg kg⁻¹ RTX, a capsaicin analog, dissolved in 1.2% DMSO with 0.06% Tween 80 in PBS, was injected subcutaneously into the flank on day −3. Control mice received an injection of vehicle solution (PBS with 1.2% DMSO and 0.06% Tween 80) on the same day. Vehicle and RTX treated mice were housed together before and during the psoriatic modeling.

## Primary culture of DRG neurons
Total DRG from 6- to 8-week-old mice were dissected and transferred to Neurobasal-A medium and digested in HEPES-buffered saline (Sigma-Aldrich, Catalog no. 51558) containing 1 mg ml⁻¹ collagenase A and 2.4 U ml⁻¹ Dispase II (Roche Applied Sciences, Catalog no. 04942078001) for 30 min at 37 °C. After dissociation, cells were washed with DMEM containing 10% fetal bovine serum (FBS). Neurons were gently triturated, pelleted, and then resuspended in Neurobasal-A culture medium (Thermo Fisher Scientific, Catalog no. 10888022) containing 2% B-27 supplement (Gibco, Catalog no. 17504044). Neurons were plated on poly-D-lysine coated sterile cell culture dishes containing Neurobasal-A medium, B-27 supplement and L-glutamine (2 mM, Gibco, Catalog no. 25030081). Half of the medium was replaced with fresh medium every 2 days. On day 7, neurons were stimulated for CGRP assays for experiments as described in neuronal stimulation and CGRP release.

## Immunofluorescence staining
For immunofluorescence staining, dorsal skin tissues and DRG tissues were dissected from mice previously euthanized and transcardially perfused with ice-cold PBS followed by 4% paraformaldehyde in PBS (PBS/4%PFA). Both skin tissues and DRG samples were post-fixed in PBS/4% PFA at 4 °C for 12 h and then cryoprotected in PBS/30% sucrose (Sigma-Aldrich, Catalog no. S0389) at 4 °C for 3 days. The tissues were then embedded in Optimal Cutting Temperature (OCT) and stored in −80 °C until processing. Frozen tissues were sliced using a microtome to 20-μm thick sections. Cryosections were blocked in blocking buffer containing PBS with 10% goat serum, 0.3% Triton X-100 and 0.5% bovine serum albumin and stained with rabbit anti-CGRP (Cell Signaling Technology, Catalog no. 14959, 1:200 diluted) or mouse anti-PGP9.5 (Abcam, Catalog no. ab8189, 1:200 diluted) over night at 4 °C. Tissue sections were then washed 3 times with staining buffer and incubated with secondary antibody at room temperature. After washing, sections were counterstained with DAPI (1 μg ml⁻¹) (Sigma-Aldrich, Catalog no. D9542) for nuclear visualization. Images were acquired on an Olympus Fluoview Confocal microscope.

## Histological analysis of skin inflammation
Skin biopsies were dissected immediately after animals were deeply anesthetized and intracardially perfused with ice cold PBS, followed by PBS/4% PFA. Inflamed dorsal skin and ears were fixed in formalin and embedded in paraffin. Sections (10 μm) were subjected to histological analysis by hematoxylin and eosin (H&E) staining. For immunohistochemical staining, the tissue sections were deparaffinized and washed in PBS. Antigen retrieval was conducted by pepsin and incubation in 3% hydrogen peroxide for 10 min at room temperature. The sections were blocked in PBS containing 1% bovine serum albumin for 1 h at room temperature, and then stained in a blocking buffer containing

anti-mouse Ki67 (ServiceBio, Catalog no. GB121141, 1:500 diluted) and anti-Cytokeratin 5 (Abcam, Catalog no. ab52635, dilution 1:500) overnight at 4 °C. On the following day, the sections were stained with an HRP-polymer complex at room temperature for 20 min and incubated with the secondary antibody for 20 min. The sections were developed with DAB reagent and then counterstained with hematoxylin. The sections were washed with tap water and increasing concentrations of ethanol for dehydration. Epidermal thickness was assessed by measuring from the basal membrane to the cornified layer of the epidermis. Three areas were randomly taken for measurement from each slice.

### Enzyme-linked immunosorbent assay

To detect murine IL-17, IL-22, and IL-23 in lesional skin, ELISA kits were used and experiments performed following the manufacturer's instructions. Briefly, 100–200 mg of mouse dorsal skin was grounded in radioimmunoprecipitation assay buffer and analyzed.

### Neuronal stimulation and CGRP release

Cultures were maintained for 7 days as described in the section primary culture of DRG neurons. On the day of experiment, the neuronal culture medium was removed from all wells and replaced with extracellular solution (ECS) containing the following (in mM): 150 NaCl, 10 HEPES, 10 glucose, 5 KCl, 2 CaCl$_2$, and 1 MgCl$_2$; adjusted to the final pH with Tris-base. Cells were incubated for 30 min at 37 °C with 30 mM KCl, acidic solution (pH 5.5), GMQ (1 mM, Sigma Aldrich, Catalog no. SML0874), acidic solution plus amiloride (100 μM, Sigma-Aldrich, Catalog no. A7410) or bicuculline (10 μM, MCE, Catalog no. 485-49-4), all diluted in ECS. The solution was collected and assayed to determine concentration with a CGRP EIA kit (Cayman Chemical, Catalog no. 589001) according to the manufacturer's instructions. Plates were read using a Synergy HTX Multi-Mode Reader at 414 nm.

### CGRP release assay from skin explants

To assess CGRP release in inflamed skin, we conducted a CGRP release assay following a previously described protocol[18]. In brief, 12-mm skin punch biopsies were obtained from both uninflamed and inflamed dorsal skin samples and placed in 24-well plates containing 1 ml DMEM. The explants were then incubated at 32 °C with gentle shaking (150 rpm) for 45 min. Following the incubation, the media from the organ cultures were collected, and CGRP levels were measured using the CGRP EIA kit as per the manufacturer's instructions.

### Virus construction and injection

The following viruses were used: pAAV-CAG-eGFP-U6-NC (Serotype PHP.S), pAAV-CAG-eGFP-U6-ASIC3-ShRNA (Serotype PHP.S) and pAAV-U6-ASIC3-ShRNA-EF1α-DIO-FLAG-ASIC3*-2A-mCherry (Serotype PHP.S). All were generated by SunBio Medical Biotechnology Co. Ltd. (Shanghai). All viral vectors were stored in aliquots at −80 °C until further use. Viral infusion was conducted via tail vein injection. The injection sites were examined by the expression of the fluorescent proteins, EGFP or mCherry. Four weeks after viral infusion, psoriatic modeling were performed.

### Real-time quantitative reverse transcription PCR (qRT-PCR) analysis

Total RNA from DRG tissues was isolated using TRIzol reagent (Invitrogen, Catalog no. 15596026) and 1 μg of total RNA was reverse-transcribed into cDNA. The relative expression of the target genes was normalized to the expression of *Gapdh* and calculated with the $2^{-\Delta\Delta Ct}$ method. The primer sequences of the target and control genes were as follows: *Asic3* forward: 5′-CAGCCCTGTGGACCTGAGAA and reverse-3′, 5′-CGCCCTTAGGAGTGGTGAGC-3′; *Gapdh* forward: 5′-CCTCGTCCCGTA GACAAAATGGT-3′ and reverse, 5′-CCTCGTCCCGTAGACAAAATGGT-3′.

### CCK-8 assay

The ability of cell proliferation was determined using the cell counting kit-8 (CCK-8) assay (Beyotime, Catalog no. C0038). Briefly, cells were planted in 96-well plates (4000 cells/well) and examined over 24 h. The CCK8 solution (10 μl) and medium (100 μl) were added into each well and the absorbance measured spectrophotometrically at 450 nm after incubation.

### Sensory neuron-BMDC coculture

Mouse bone marrow-derived dendritic cells (BMDCs) were prepared from bone marrow cells extracted from the femurs and tibias of mouse. The cells were cultured in RPMI-1640 medium containing 10% fetal bovine serum, 1% antibiotic-antimycotic, 20 ng/ml GM-CSF (PeproTech, Catalog No. 214-14) and 10 ng/ml IL-4 (PeproTech, Catalog No. 315-03). The primary DRG neurons were prepared as illustrated before and cultured at the density of $1.5 \times 10^4$ cells/well in 24-well plates for 3 days. Harvested BMDCs were resuspended on day 7 and seeded on top of the DRG neurons at the density of $5 \times 10^5$ cells/well. The co-incubated cells were treated with pH 5.5 or 1 μg/ml lipopolysaccharide and R848 or vehicle control with BIBN4096 for 24 h. BMDCs were then dissociated with 0.25% trypsin-EDTA and collected for flow cytometry.

### Flow cytometry

BMDCs were surface-stained with Alexa Fluor 700 anti-mouse CD45 antibody (BioLegend, Catalog no. 103128, 1:200 diluted), PE-Cyanine 7-conjugated CD11c antibody (Invitrogen, Catalog no. 25-0114-82, 1:200 diluted) and FITC-conjugated major histocompatibility complex II antibody (Invitrogen, Catalog no. 11-5321-85, 1:200 diluted). Cells were then fixed and permeated and stained with PerCP-conjugated IL-23 antibody (Invitrogen, Catalog no. 46-7023-82, 1:100 diluted). Data were obtained using LSRFortessa (BD Biosciences) and analyzed with FlowJo.

### Skin pH assessment

The skin surface pH was detected using an electrochemical sensor developed by Prof Yang Tian[40]. Briefly, Hemin-aminoferrocene (Hemin-Fc) complexes were synthesized, which linked a hemoglobin unit to two aminoferrocene molecules via amide bonds. Then the Hemin-Fc was stably immobilized on the surface of the 7 μm carbon fiber electrode by π-π stacking. As the pH decreased from 8.0 to 5.5, the E1/2 value of $Fe^{2+/3+}$ in hemoglobin, which was located at −356 mV (half-wave potential E1/2 with respect to Ag/AgCl), shifted towards the positive direction. This made it possible to reflect changes in pH by monitoring changes in the potential of Hemin-Fc.

Mice were anesthetized by induction with 3% isoflurane, and anesthesia was maintained with 1.5% isoflurane afterwards. The animal was fixed by placing its head/nose in an anesthesia mask and prone on the stereotactic machine (RWD Life Science, Catalog no. 68026). Mouse coat was wetted using a gauze moistened with sterile water and shaved with a razor in the direction of the coat. Afterwards, the microelectrodes were placed close to the skin surface for pH determination. The level of pH was detected and quantified using an in-house computer program based on the functions spline and splint.

### Lipid extraction and lipid chromatography-mass spectrometry (LS-MS) analysis

Briefly, lipids were extracted according to the MTBE method. The analysis was performed on a UHPLC system (LC-30AD, Shimadzu) coupled with QTRAP MS (6500+, Sciex). The analytes were separated on HILIC (Phenomenex, Luna NH2, 2.0 mm × 100 mm, 3 μm) and C18 column (Phenomenex, Kinetex C18, 2.1 × 100 mm, 2.6 μm). Sciex OS was used for quantitative data processing.

## Patch clamp experiments

The whole-cell configuration of the patch clamp technique was used to record membrane currents (voltage clamp). Cells were bathed in extracellular fluid containing (in mM): 150 NaCl, 5 KCl, 2 $CaCl_2$, 1 $MgCl_2$, 10 glucose, and 10 HEPES, pH 6.0–7.4 adjusted with Tris-base. The pipette solution (internal) contained (in mM): 120 KCl, 30 NaCl, 1 $MgCl_2$, 0.5 $CaCl_2$, 5 EGTA, 2 MgATP, 0.3 $Na_2GTP$, and 10 HEPES, pH 7.4 adjusted with Tris-base. Cells were held at −60 mV in whole-cell mode and currents were digitized at 10 kHz and filtered at 2 kHz. All data were acquired in the voltage-clamp mode using Axon Digidata 1550B and MultiClamp 700B amplifier.

## Statistics and reproducibility

The experimenters were blind to group allocation during data collection to avoid experimenter bias. All data presented in the work were obtained from at least three biological replicates independently. All summary data are presented as the mean ± SEM. Most histograms display individual data points that represent the values and numbers of individual samples for each condition. Data distributions were tested for normality and the homogeneity of variance among groups was determined using the Levene's test. Data were analyzed with unpaired Student's $t$ test, one-way analysis of variance (ANOVA) or two-way ANOVA. For post hoc analysis, we used Bonferroni's corrections for multiple comparisons. Statistical analyses were performed with GraphPad Prism (version 8.0.2, GraphPad Software, Inc., USA). $p$ value < 0.05 was considered statistically significant. Statistical significance is mainly displayed as $*p < 0.05$, $**p < 0.01$, and $***p < 0.001$, and in some cases is indicated as $^{\#}p < 0.05$, $^{\#\#}p < 0.01$, and $^{\#\#\#}p < 0.001$ for multiple comparisons. Not significant values are not denoted except for emphasis.

## Reporting summary

Further information on research design is available in the Nature Portfolio Reporting Summary linked to this article.

## Data availability

All data supporting the findings of this study are available within the paper and its Supplementary Information. All raw mass spectrometry data can be found in the MetaboLights Database (https://www.ebi.ac.uk/metabolights/) under the accession code MTBLS10273. Source data are provided with this paper.

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

## Acknowledgements

We thank Dr. Honglin Wang's group at Shanghai Jiao Tong University for their technique assistance. We thank Drs John A. Wemmie, Margaret P. Price, Michael J. Welsh and Chih-Cheng Chen for kindly providing the *Asic3* KO and *Asic3^flox/flox* mice used in the current study. We also thank Profs. Rohini Kuner and Xu Zhang for their kind permission to use and for providing the *Na_V1.8^Cre* mice used in the current study. This study was supported by grants from the STI2030-Major Projects (2021ZD0202800), the National Natural Science Foundation of China (81961128024, 31930050 and 32300789), the National Institutes of Health (NS114716), the Science and Technology Commission of Shanghai Municipality (22XD1420700), the Shanghai Municipal Health Commission (202040015 and 2022XD046), and innovative research team of high-level local universities in Shanghai.

## Author contributions

C.H., P.-Y.S., X.Q., W.-G.L., and T.-L.X. conceived the project, designed the experiments, and interpreted the results. C.H., P.-Y.S., Y.J., and X.Q. performed the majority of mouse modeling, immunohistochemistry, genetic and pharmacological intervention and data analysis. Q.J. and Y.L. performed mouse genotyping. S.-L.H., B.-S.W., Y.-X.H., and A.-R.R.

assisted with the animal experiments and data analysis. X.Q. and J.-F.L. contributed to construction of ASIC3 reporter mice, immunohistochemistry and data analysis. C.H., Y.L., Z.L., and Y.T. performed skin pH measurement. C.H., P.-Y.S., and X.Q. performed lipid metabolomics analysis. C.H., P.-Y.S., Z.Y., M.X.Z., X.Q., W.-G.L., and T.-L.X. wrote the manuscript with contributions from all the authors. All the authors approved the final version of the manuscript.

## Competing interests

The authors declare no competing interests.
