## [Peer Review File · Nature Communications]

Sensory ASIC3 channel exacerbates psoriatic inflammation via a neurogenic pathwayREVIEWER COMMENTS

Reviewer #1 (Remarks to the Author):

This is an interesting study to show a role of sensory neuron ASIC3 in psoriatic inflammation in mouse models. The authors have used ASIC3 knockout (and conditional KO), ASIC3 knockdown and gene rescue approaches to validate the involvement of sensory neuron ASIC3 in IMQ-induced type 17 inflammation. The authors further demonstrated the neuron ablation of TRPV1-expressing afferents (by RTX) and inhibiting CGRP release from skin afferents (by botulinum neurotoxin A) can result in similar effects to ASIC3 knockout (or knockdown) to attenuate psoriatic inflammation. Also, pharmacological blockade of CGRP signaling (by BIBN4096) attenuated IMQ-induced inflammation in ASIC3 wild-type mice, whereas CGRP pretreatment rescued the IMQ-induced inflammation phenotypes in ASIC3 knockout mice. However, although much effort has proved a role for ASIC3 involved in IMQ-induced psoriatic inflammation, no detail mechanistic insight had offered to advance our understanding how the ASIC3-mediated neurogenic effects are activated and how this ASIC3 signaling is different from the TRPV1 pathway in the context of IMQ-induced psoriatic inflammation. Beneath are some specific comments.

1. The novelty of this study is limited without knowing the ASIC3-expressing neuron types and how ASIC3 of skin afferents is activated in the context of IMQ-induced inflammation. Also, how CGRP signaling can lead to abnormal epidermal proliferation and up-regulation of type 17 cytokines should be addressed.
2. In general, the statistical analyses are not appropriate. In most 2x2 experimental groups, 2-way ANOVA instead of one-way ANOVA should be used.
3. Baseline data (vehicle control) of epidermal thickness, IL-17, IL-22, and IL-23 are missing in the Figure 1.
4. In Figure 2a, the validation of Nav1.8 conditional ASIC3 knockout should be analyzed by a quantitative approach (e.g., QPCR).
5. How are ASIC3 and TRPV1 co-localized in skin afferents innervating dorsal skin?
6. Is the IL-23-induced psoriatic inflammation (on ears) a relevant model with IMQ-induced one (on dorsal skin)? The rationale of the IL-23-induced psoriatic inflammation experiment is not clear. It may be more important to know whether IL-23 is a downstream signaling of ASIC3 activation and/or CGRP receptors.
7. The Figure 3b does not support that IL-23 treatment can induce splenomegaly.
8. It is surprising to see KCL-induced CGRP release is ASIC3-dependent in cultured DRG neurons (Fig. 5a). Of note, GMQ also induced CRGP release in ASIC3 knockout. What are the interpretations of these data?
9. The rationale to measure CGRP in the skin explant is not clear. The different CGRP levels between normal and IMQ-treated skin cannot tell the contribution of neurogenic effect from skin afferents.
10. The data of Figure 5c is not convincing. Quantitative analysis is needed.
11. It is not clear to what extent the ASIC3-expressing skin afferents could contribute to the IMQ-induced psoriatic inflammation.
12. The discussion section should be more carefully organized. There is too much speculation in the discussion.
13. Were all studied performed in male or female mice (or both)? What are the mouse ages?
14. What were the efficacies of ASIC3 knockdown and gene rescue in the supplementary Figure 2?
15. How were the skin samples collected and processed for ELISA assays? Especially, it is important to know how the lesional skin is defined.

Reviewer #2 (Remarks to the Author):

The current study highlights the importance of ASIC3, an ion channel previously overlooked in skin inflammation but known for its involvement in itch and nociception. The investigators found that ASIC3 contributes to the psoriatic skin phenotype and accompanying type 17 inflammation. Knocking out *Asic3* globally or specifically in nociceptors in mice significantly reduced psoriatic acanthosis and type 17 inflammation induced by IMQ, comparable to nociceptor ablation. However,

ASIC3 was not essential for IL-23-induced psoriatic inflammation, which can bypass the need for nociceptors. Further investigation revealed that ASIC3 activation leads to activity-dependent release of CGRP from sensory neurons. By using botulinum neurotoxin A and CGRP antagonists, they were able to prevent the sensory neuron-mediated exacerbation of psoriatic inflammation, similar to the effects observed with Asic3 knockout. Conversely, replenishing CGRP in Asic3 knockout skin restored the inflammatory response. The study establishes sensory ASIC3 as a critical component in psoriatic inflammation and a promising target for managing chronic skin diseases.

While the study is potentially interesting, there are several results that do not support the conclusion they have made. Additionally, there are several important experiments that are missing.

From the experiment shown in Figure 3b they claim IL-23 administration induced significant splenomegaly in both groups. However, the experiment does not have any vehicle control. As compared to the vehicle controls in Figure 1b (~70 mg), it does not have significant increase in spleen weight after IL-23 intradermal administration in both WT and KO in Figure 3b (also around 70 mg).

Are there any significant increases in Ki67 positive cells in WT and KO mice after IL-23 treatment as compared to PBS controls (Figure 3d)? If not, it suggests IL-23 treatment does not induce significant cell proliferation. In this study, they need to specify the type of cells that Ki67 positive cells represent.

It is unclear whether BoNT/A can directly block CGRP release which is an important point for the study.

Another important point remains unaddressed is what activate ASIC3 in psoriatic skin. Can they detect acidification or LPC increase in psoriatic skin?

In Figure 5a,b, GMQ seems also increase CGRP release in ASIC3 KO DRG. If this is the case, it suggests ASIC3 is not the only mediator to induce CGRP release.

The quality of nerve image in Fig 5C is poor. It is hard to see any nerve staining. They should include negative control for CGRP staining. Also, they need to quantify the nerve counts in order to make the conclusion. CGRP injection increased only IL17 but not other cytokines. So CGRP itself does not induce a full-scale inflammatory responses.

Responses to Reviewers' comments (Reviewers' comments in *italic*)

Reviewer #1:

This is an interesting study to show a role of sensory neuron ASIC3 in psoriatic inflammation in mouse models. The authors have used ASIC3 knockout (and conditional KO), ASIC3 knockdown and gene rescue approaches to validate the involvement of sensory neuron ASIC3 in IMQ-induced type 17 inflammation. The authors further demonstrated the neuron ablation of TRPV1-expressing afferents (by RTX) and inhibiting CGRP release from skin afferents (by botulinum neurotoxin A) can result in similar effects to ASIC3 knockout (or knockdown) to attenuate psoriatic inflammation. Also, pharmacological blockade of CGRP signaling (by BIBN4096) attenuated IMQ-induced inflammation in ASIC3 wild-type mice, whereas CGRP pretreatment rescued the IMQ-induced inflammation phenotypes in ASIC3 knockout mice.

We thank the reviewer for the positive comments.

However, although much effort has proved a role for ASIC3 involved in IMQ-induced psoriatic inflammation, no detail mechanistic insight had offered to advance our understanding how the ASIC3-mediated neurogenic effects are activated and how this ASIC3 signaling is different from the TRPV1 pathway in the context of IMQ-induced psoriatic inflammation.

In response to the insightful feedback from the reviewer, we have performed additional experiments, incorporated new data, and provided expanded explanations in the Discussion section. Specifically, we have addressed the mechanisms triggering ASIC3-mediated neurogenic effects (please see Fig. 7, Supplementary Figs. 10 and 11, and their legends, page 15, the last paragraph to page 16, the 1st paragraph) and elucidated the distinctions between ASIC3 signaling and the TRPV1 pathway within the context of IMQ-induced psoriatic inflammation (please see Supplementary Fig. 5 and its legend, page 9, end of the

last paragraph to page 10, beginning of the 1st paragraph).

Beneath are some specific comments.

1. The novelty of this study is limited without knowing the ASIC3-expressing neuron types and how ASIC3 of skin afferents is activated in the context of IMQ-induced inflammation. Also, how CGRP signaling can lead to abnormal epidermal proliferation and up-regulation of type 17 cytokines should be addressed.

In response to the reviewer's suggestion, we have integrated new data obtained from a recently established ASIC3-myc reporter mouse model. This includes a comprehensive characterization of ASIC3 distribution specifically in TRPV1⁺ neurons and skin afferents (please see Supplementary Fig. 5 and its legend, page 9, end of the last paragraph to page 10, beginning of the 1st paragraph).

Furthermore, we have included novel findings showing skin acidosis and increased presence of LPC in IMQ-induced psoriatic lesions (please see Fig. 7, Supplementary Figs. 10 and 11, and their legends, page 15, the last paragraph to page 16, the 1st paragraph). These insights shed light on potential mechanisms of ASIC3 activation within the context of IMQ-induced inflammation.

In addition, we have conducted additional experiments to specifically address how CGRP signaling is involved in abnormal epidermal proliferation and up-regulation of type 17 cytokines (please see Fig. 7, Supplementary Figs. 8 and 9, and their legends, page 14, the last paragraph to page 15, the 2nd paragraph).

2. In general, the statistical analyses are not appropriate. In most 2x2 experimental groups, 2-way ANOVA instead of one-way ANOVA should be used.

Corrected accordingly (please see the Figure legends).

3. *Baseline data (vehicle control) of epidermal thickness, IL-17, IL-22, and IL-23 are missing in the Figure 1.*

Included as suggested (please see new Fig. 1).

4. *In Figure 2a, the validation of Nav1.8 conditional ASIC3 knockout should be analyzed by a quantitative approach (e.g., QPCR).*

Done accordingly (please see new Supplementary Fig. 2).

5. *How are ASIC3 and TRPV1 co-localized in skin afferents innervating dorsal skin?*

Done accordingly in an ASIC3-myc reporter mice (please see Supplementary Fig. 5 and its legend, page 9, end of the last paragraph to page 10, beginning of the 1st paragraph).

6. *Is the IL-23-induced psoriatic inflammation (on ears) a relevant model with IMQ-induced one (on dorsal skin)? The rationale of the IL-23-induced psoriatic inflammation experiment is not clear. It may be more important to know whether IL-23 is a downstream signaling of ASIC3 activation and/or CGRP receptors.*

Yes. The IL-23-induced psoriatic inflammation (on ears) has been demonstrated to be a pertinent model in conjunction with the IMQ-induced model (on dorsal skin). We have clarified the rationale for the IL-23-induced psoriatic inflammation experiment (please see page 11, beginning of the 1st paragraph).

In response to the reviewer's suggestion, we have conducted additional experiments utilizing a co-culture system involving DRG neurons and bone marrow-derived DCs (BMDCs). These experiments demonstrate that IL-23 is indeed a downstream signaling molecule produced in response to acid stimulation

of ASIC3 activation and the subsequent activation of CGRP receptors (please see new Supplementary Fig. 8, page 14, the last paragraph to page 15, the 1st paragraph).

7. The Figure 3b does not support that IL-23 treatment can induce splenomegaly.

Yes. The lack of splenomegaly observed after local IL-23 injection in both groups suggests the absence of systemic effects (please see new Fig. 3b, page 11, middle of the 1st paragraph).

8. It is surprising to see KCl-induced CGRP release is ASIC3-dependent in cultured DRG neurons (Fig. 5a). Of note, GMQ also induced CRGP release in ASIC3 knockout. What are the interpretations of these data?

We appreciate the reviewer for bringing this to our attention. After a thorough reassessment of the experiments, we identified an error in the previous version of our manuscript regarding the interpretation of the KCl-induced CGRP release in DRG neurons. We have rectified this in the revised manuscript, clarifying that high potassium (30 mM KCl) induces a significant increase in CGRP secretion from both *Asic3*^{+/+} and *Asic3*^{-/-} DRG neurons (please see new Fig. 5b).

Moreover, regarding the observation that GMQ induced CGRP release in *Asic3* knockout mice, it is essential to note that GMQ has multiple molecular targets, including acting as a nonproton agonist for ASIC3 (Ref. 1) and as an antagonist for GABA_A receptors (Ref. 2). The effects of GMQ in ASIC3 knockout are likely attributed to its influence on GABA_A receptors. Consistently, another GABA_A receptor antagonist, bicuculline also induced CGRP release in an ASIC3-independent manner (please see new Supplementary Fig. 7b and its legend, page 13, middle of the 1st paragraph).

References:

1. Yu Y, Chen Z, Li WG, Cao H, Feng EG, Yu F, Liu H, Jiang H, Xu TL. A nonproton ligand sensor in the acid-sensing ion channel. *Neuron* 2010; 68(1): 61-72.
2. Xiao X, Zhu MX, Xu TL. 2-Guanidine-4-methylquinazoline acts as a novel competitive antagonist of A type γ -aminobutyric acid receptors. *Neuropharmacology* 2013; 75: 126-37.

9. *The rationale to measure CGRP in the skin explant is not clear. The different CGRP levels between normal and IMQ-treated skin cannot tell the contribution of neurogenic effect from skin afferents.*

The primary reason for assessing CGRP levels in the skin explant was to investigate local CGRP release in the context of IMQ-induced psoriatic inflammation. We sought to identify differences in CGRP levels between normal and IMQ-treated skin, with an intention to establish a potential contribution of neurogenic effects from skin afferents to the observed differences. We acknowledge that these data alone do not definitively pinpoint to the specific origin of CGRP in the skin explant.

However, in conjunction with other complementary approaches, including the assessment of CGRP levels in cultured DRG neurons and the measurement of CGRP⁺ and PGP9.5⁺ nerve densities in psoriatic lesions, the observed changes in CGRP levels in the IMQ-treated skin compared to the normal skin explant align with the influence of neurogenic effects from skin afferents. This reinforces the proposition that neurogenic CGRP is a pivotal driver of psoriatic inflammation.

10. *The data of Figure 5c is not convincing. Quantitative analysis is needed.*

Done accordingly (please see new Fig. 5e).

11. *It is not clear to what extent the ASIC3-expressing skin afferents could contribute*

to the IMQ-induced psoriatic inflammation.

We appreciate the reviewer's point regarding the necessity to clarify the extent by which ASIC3-expressing skin afferents contribute to IMQ-induced psoriatic inflammation. Our results demonstrate that *Asic3* KO mice exhibit a comparable improvement in psoriatic inflammation as those with their TRPV1⁺ nociceptors ablated. This suggests that ASIC3-expressing skin afferents play a role comparable to nociceptive sensory neurons in driving psoriatic inflammation.

12. The discussion section should be more carefully organized. There is too much speculation in the discussion.

In response to the reviewer's valuable feedback, we have reorganized and refined the Discussion section to enhance clarity and reduce speculative elements. We have also incorporated insights gained from our new experiments (please see new Discussion section).

13. Were all studies performed in male or female mice (or both)? What are the mouse ages?

The information is now provided in the revised manuscript (please see page 22, middle of the 1st paragraph).

14. What were the efficacies of ASIC3 knockdown and gene rescue in the supplementary Figure 2?

New figures to quantify the efficacies of ASIC3 knockdown and gene rescue are now provided in the revised manuscript (please see new Supplementary Fig. 3 and Supplementary Fig. 4).

15. How were the skin samples collected and processed for ELISA assays? Especially, it is important to know how the lesional skin is defined.

Six hours after the final IMQ treatment, we harvested a 12 mm punch biopsy from the treated skin, which was then placed in a 24-well plate containing 1 ml DMEM. The samples were incubated in a 32 °C shaking incubator (150 rpm) for 45 min, and the media were collected for analysis using the CGRP EIA kit (Cayman Chemical Co.) following the procedures outlined in the literature¹ (please see the section of CGRP release assay from skin explants in Methods).

It is crucial to note that the punch biopsies were obtained from either non-lesioned or lesioned skin, where the sample corresponds to the area of the dorsal skin treated with either vehicle or IMQ, as confirmed by H & E staining. No lesioned skin areas were observed in vehicle-treated samples.

References:

1. Cohen JA, Edwards TN, Liu AW, Hirai T, Jones MR, Wu J, Li Y, Zhang S, Ho J, Davis BM, Albers KM, Kaplan DH. Cutaneous TRPV1⁺ neurons trigger protective innate type 17 anticipatory immunity. *Cell*. 2019; 178(4): 919-932.e14.

Reviewer #2:

The current study highlights the importance of ASIC3, an ion channel previously overlooked in skin inflammation but known for its involvement in itch and nociception. The investigators found that ASIC3 contributes to the psoriatic skin phenotype and accompanying type 17 inflammation. Knocking out Asic3 globally or specifically in nociceptors in mice significantly reduced psoriatic acanthosis and type 17 inflammation induced by IMQ, comparable to nociceptor ablation. However, ASIC3 was not essential for IL-23-induced psoriatic inflammation, which can bypass the need for nociceptors. Further investigation revealed that ASIC3 activation leads to activity-dependent release of CGRP from sensory neurons. By using botulinum neurotoxin A

and CGRP antagonists, they were able to prevent the sensory neuron-mediated exacerbation of psoriatic inflammation, similar to the effects observed with Asic3 knockout. Conversely, replenishing CGRP in Asic3 knockout skin restored the inflammatory response. The study establishes sensory ASIC3 as a critical component in psoriatic inflammation and a promising target for managing chronic skin diseases. While the study is potentially interesting, there are several results that do not support the conclusion they have made. Additionally, there are several important experiments that are missing.

We appreciate the positive feedback from the reviewer. Additionally, following the reviewer's suggestions, we conducted additional experiments.

1. From the experiment shown in Figure 3b they claim IL-23 administration induced significant splenomegaly in both groups. However, the experiment does not have any vehicle control. As compared to the vehicle controls in Figure 1b (~70 mg), it does not have significant increase in spleen weight after IL-23 intradermal administration in both WT and KO in Figure 3b (also around 70 mg).

We apologize for any confusion caused by the previous version of our manuscript. In response to the reviewer's comment, we administered IL-23 (1 µg) by injection into one ear every other day for a total of 8 days. The results indicate that the IL-23 injection did not induce splenomegaly in either *Asic3*^{+/+} or *Asic3*^{-/-} mice (compared to the vehicle control groups, please see new Fig. 3b and its legend, page 11, middle of the 1st paragraph).

2. Are there any significant increases in Ki67 positive cells in WT and KO mice after IL-23 treatment as compared to PBS controls (Figure 3d)? If not, it suggests IL-23 treatment does not induce significant cell proliferation. In this study, they need to specify the type of cells that Ki67 positive cells represent.

We have modified the IL-23 injection dosage following the protocols outlined in the literature¹. With this adjustment, IL-23 treatment resulted in a significant increase in cell proliferation, as evidenced by Ki67 immunohistochemical staining (please see new Fig. 3e,f and its legend, page 11, the 1st paragraph).

Furthermore, we examined the cell type(s) of Ki67-positive cells, as depicted in the new Supplementary Fig. 1. Many of the Ki67-positive cells in the psoriatic models were co-labeled with Keratin 5 (Krt5), a cytoskeleton marker typically expressed in the basal and spinous epidermal layers.

References:

1. Wu R, Zeng J, Yuan J, Deng X, Huang Y, Chen L, Zhang P, Feng H, Liu Z, Wang Z, Gao X, Wu H, Wang H, Su Y, Zhao M, Lu Q. MicroRNA-210 overexpression promotes psoriasis-like inflammation by inducing Th1 and Th17 cell differentiation. *J Clin Invest.* 2018; 128(6): 2551-2568.
3. *It is unclear whether BoNT/A can directly block CGRP release which is an important point for the study.*

We have investigated whether BoNT/A blocks acid-induced CGRP release in cultured DRG neurons, and found this indeed is the case (please see new Supplementary Fig. 7a and its legend).

4. *Another important point remains unaddressed is what activate ASIC3 in psoriatic skin. Can they detect acidification or LPC increase in psoriatic skin?*

Following the reviewer's suggestion, we have included new data showing skin acidosis and augmentation of LPC levels in IMQ-induced psoriatic lesions (please see Fig. 7, Supplementary Figs. 10 and 11, and their legends, page 15, the last

paragraph to page 16, the 1st paragraph). These insights shed lights on potential mechanisms of ASIC3 activation within the context of IMQ-induced inflammation.

5. In Figure 5a,b, GMQ seems also increase CGRP release in ASIC3 KO DRG. If this is the case, it suggests ASIC3 is not the only mediator to induce CGRP release.

Regarding the observation that GMQ induced CGRP release in *Asic3* knockout mice, it is essential to note that GMQ has multiple molecular targets, including acting as a nonproton agonist of ASIC3 and as an antagonist of GABA_A receptors. The effects of GMQ on ASIC3 knockout could be attributed to its influence on GABA_A receptors. Supporting this view, another GABA_A receptor antagonist, bicuculline, also induced CGRP release in an ASIC3-independent manner (please see **new Supplementary Fig. 7b and its legend, page 13, middle of the 1st paragraph).**

6. The quality of nerve image in Fig 5C is poor. It is hard to see any nerve staining. They should include negative control for CGRP staining. Also, they need to quantify the nerve counts in order to make the conclusion. CGRP injection increased only IL17 but not other cytokines. So CGRP itself does not induce a full-scale inflammatory responses.

We apologize for the previous image quality issue and appreciate the reviewer's feedback. To address this concern, we have conducted additional immunofluorescence staining for both CGRP and PGP9.5 and obtained nerve images with enhanced quality. Furthermore, we have quantified nerve counts to provide a more comprehensive analysis (please see **Fig. 5d,e and its legend).**

Regarding the impact of CGRP injection on cytokine release, we reevaluated our data. By directly comparing the differences between the vehicle and CGRP-treated groups in *Asic3*^{-/-} mice using Student's *t*-test, we found that CGRP

injection increased IL-17, IL-22, and IL-23 cytokines (please see new Fig. 6e and its legend). This suggests that CGRP itself induces a full-scale inflammatory response.

REVIEWER COMMENTS

Reviewer #1 (Remarks to the Author):

The authors have addressed most of my comments and enthusiastically added new data to fill up the knowledge gaps regarding how ASIC3-mediated neurogenic effects are activated and involved in IMQ-induced psoriatic inflammation. Still, there are some issues need to be further clarified.

1. What is the Y-axis unit for Ki67 data in Figure 1d, 2e, Supplementary Figure 3j, 4j?
2. The IL17 data between Figure 1e and Figure 2f are quite different. What is the explanation?
3. Issues of two-way ANOVA in Figure 3b (and other Figures) are identified. What is the F value?
4. In Figure 5, please define the extracellular solution (ECS) group. Does ECS mean the vehicle control? Or anything else?
5. The Figure 5b looks like a new data set that the authors had corrected from the previous mistake. Of note, the values of ECS groups are very different between Figure 5a and Figure 5b. Why?
6. Detail information of Supplementary Figure 5 is missing. (a,b) What are the primers 1 and primers 2? (d) Which skin was sampled? What does "relative images" mean? What do arrows indicate? Are those images covered dermis or epidermis or both? How can it be sure that the ASIC3-myc+ (green) signals are sensory nerves?
7. The involvement of LPC14:0 in activation of ASIC3-mediated neurogenic effect in the context of psoriatic inflammation needs further clarification. LPC14:0 is a relatively rare LPC species in blood samples. It looks like the amount of LPC14:0 is also low in skin samples (Supplementary Figure 11). The amount of LPC14:0 in blood samples is 10-100 folds lower than common LPC species like LPC16:0, LPC18:0, LPC18:1. It is hardly convincing that ~2 folds increase of LPC14:0 in skin can have a dramatically biological effect, while a high concentration (10 μ M) of LPC14:0 is required to potentiate acid-induced current via ASIC3. Of note, LPC16:0 and LPC18:1 (1-10 μ M) can directly activate ASIC3 in neutral pH in whole-cell patch clamp recordings. What are the ratios of LPC14:0 vs. LPC18:1 in normal and lesion skin? In theory, LPC18:1 would have better chance to dominate the ASIC3 activation, but lesion skin did not show higher levels of LPC18:1 than normal skin.
8. In Figure 7d, pH7.4 + LPC14:0 can trigger CGRP release via an ASIC3-dependent manner. Can LPC14:0 directly activate ASIC3 at neutral pH?
9. Is LPC16:0 detectable in skin samples?

Minor points:

1. In page 9, line 179. The sentence is confusing. The data were compared between conditional rescue of ASIC3 in Nav1.8-Cre mice and ASIC3 knockdown mice (but not WT mice).
2. In page 22, line 461. Is it exon 11? Or exon 1?
3. A potential ethical issue is concerned. The Nav1.8-Cre mice were generated by Dr. Rohini Kuner's group. It is not appropriate to use the mice from Dr. Xu Zhang without acknowledgement of the original authors.
4. The dose of Bicuculine is missing in the Methods (and Figure legends).
5. In Supplementary Figure 10a, is the data only obtained from lesion skin? How about normal skin? How can the -0.55V be converted into pH values?

Reviewer #2 (Remarks to the Author):

The authors have nicely addressed all the reviewer's comments with new experiments. The revised manuscript is suitable for publication at Nature Communications.

Responses to Reviewers' comments (Reviewers' comments in *italic*)

Reviewer #1:

The authors have addressed most of my comments and enthusiastically added new data to fill up the knowledge gaps regarding how ASIC3-mediated neurogenic effects are activated and involved in IMQ-induced psoriatic inflammation. Still, there are some issues need to be further clarified.

We appreciate the positive feedback from the reviewer. Additionally, following the reviewer's suggestions, we conducted additional experiments and provided extensive explanations in the revised manuscript.

Beneath are some specific comments.

1. *What is the Y-axis unit for Ki67 data in Figure 1d, 2e, Supplementary Figure 3j, 4j?*

The information is now provided in the revised manuscript (please see new Figs. 1d, 2e, 3f, 4d, 6d, and 7g and new Supplementary Figs. 3j, 4g, and 6d).

2. *The IL17 data between Figure 1e and Figure 2f are quite different. What is the explanation?*

The background of mice in Fig. 1e is different from that in Fig. 2f. The control group in Fig. 1e consists of WT mice bred from the strain of *Asic3*^{+/-}, while the control group in Fig. 2f consists of *Asic3*^{flox/flox} mice from Professor Chen's lab. In Fig. 1e, the IL-17 concentration in IMQ-treated WT mice is 132.4 ± 2.4 pg/mL, while in Fig. 2f, it is 104.8 ± 12.2 pg/mL in IMQ-treated *Asic3*^{flox/flox} mice. In Fig. 1e, the IL-17 concentration in IMQ-treated *Asic3*^{-/-} mice is 81.1 ± 7.0 pg/mL, while in Fig. 2f, it is 57.3 ± 3.7 pg/mL in Nav1.8Cre::*Asic3*^{flox/flox} mice. In both cohorts, the results clearly show that ASIC3 knockout, global or selective deletion in nociceptors, downregulates IL-17 expression in psoriatic inflammation.

3. *Issues of two-way ANOVA in Figure 3b (and other Figures) are identified. What is the F value?*

Done accordingly (please see the Figure legends).

4. *In Figure 5, please define the extracellular solution (ECS) group. Does ECS mean the vehicle control? Or anything else?*

The extracellular solution (ECS) for cell culture contains the following (in mM): 150 NaCl, 10 HEPES, 10 glucose, 5 KCl, 2 CaCl₂, and 1 MgCl₂ (please refer to the Methods section for more details). ECS was used as control in Fig. 5a,b. Elsewhere in this figure, the vehicle control is the Vaseline cream applied to the dorsal skin of mice as a comparison to IMQ.

5. *The Figure 5b looks like a new data set that the authors had corrected from the previous mistake. Of note, the values of ECS groups are very different between Figure 5a and Figure 5b. Why?*

Thanks for bringing up this point. The reason for the apparent difference in the values of the ECS groups shown in Figs. 5a and 5b could be due to the varying density of DRG neurons. In each experiment, DRG neurons were seeded in 24-well plates at a density of 2×10^4 cells per well. After 7 days of culture, the alive DRG neurons often varied among different batches since the fragmented fibers and glial cells were also counted at the time of seeding. To increase the reliability of our findings, we have conducted additional experiments and updated the results as shown in new Fig. 5b and Supplementary Fig. 7b.

6. *Detail information of Supplementary Figure 5 is missing. (a,b) What are the primers 1 and primers 2? (d) Which skin was sampled? What does “relative images” mean? What do arrows indicate? Are those images covered dermis or epidermis or*

both? How can it be sure that the ASIC3-myc+ (green) signals are sensory nerves?

The sequences of primers 1 and 2 are provided in the legend of Supplementary Fig. 5 as requested. The sampled dorsal skin images covered both the dermis and epidermis. The arrows indicate the colocalization of ASIC3-myc with either TRPV1 or CGRP.

In the legend of Supplementary Figure 5, the term 'relative' has been corrected to 'representative' as appropriate.

In regard to the concern on ASIC3-myc+ (green) signals being sensory nerves, we used samples from WT mice as a negative control to verify the specificity of the myc antibody. We did not observe any significant colocalization of green (myc) and red immunostaining signals (CGRP or TRPV1) in the DRG or skin tissues (please see Supplementary Fig. 5e,f). This demonstrates that the co-localized green and red signals in sensory neurons and afferents shown in Supplementary Figure 5c and 5d do represent the expression of ASIC3-Myc in sensory neurons, although further investigation is needed to determine if any other cells in the skin also express ASIC3.

7. *The involvement of LPC14:0 in activation of ASIC3-mediated neurogenic effect in the context of psoriatic inflammation needs further clarification. LPC14:0 is a relatively rare LPC species in blood samples. It looks like the amount of LPC14:0 is also low in skin samples (Supplementary Figure 11). The amount of LPC14:0 in blood samples is 10-100 folds lower than common LPC species like LPC16:0, LPC18:0, LPC18:1. It is hardly convincing that ~2 folds increase of LPC14:0 in skin can have a dramatically biological effect, while a high concentration (10 μ M) of LPC14:0 is required to potentiate acid-induced current via ASIC3. Of note, LPC16:0 and LPC18:1 (1-10 μ M) can directly activate ASIC3 in neutral pH in whole-cell patch clamp recordings. What are the ratios of LPC14:0 vs. LPC18:1 in*

normal and lesion skin? In theory, LPC18:1 would have better chance to dominate the ASIC3 activation, but lesion skin did not show higher levels of LPC18:1 than normal skin.

We have carefully considered the reviewer's feedback and thoroughly reevaluated the lipid metabolome data. Indeed, LPC 14:0 (528 ± 48 ng/g) is less abundant than those mentioned by the reviewer, 16:0 ($37,528 \pm 3,452$ ng/g), 18:0 ($32,800 \pm 3,393$ ng/g), and 18:1 ($14,133 \pm 1,181$ ng/g). Moreover, by mainly including the molecular species of LPC that displayed significant changes in the psoriasis samples in the original manuscript, we might have given the wrong impression that perhaps all LPC species were more or less increased by the IMQ treatment. Therefore, we have reanalyzed the data by including all detected LPC molecular species (18 total) using multiple comparisons with Benjamini-Hochberg adjustment. This revealed that LPC14:0, 16:1, 18:2, 18:3, and 20:3 were significantly increased, while all others (13 total, or more than 2/3 of the species) were not significantly changed in the IMQ-treated skin samples (new Supplementary Fig. 12a). To highlight the large differences in the levels of various LPC species, we also displayed raw lipidomic data, after probabilistic quotient normalization and autoscaling, in Supplementary Fig. 12a.

Considering the reported function of the major LPC species on ASIC3 (*EMBO J*, 2016; *Pain*, 2022), we have also conducted additional experiments to determine if LPC 16:0, 18:0, and 18:1 could increase CGRP release from cultured DRG neurons. Our new data show that LPC 18:0 and 18:1, but not LPC 16:0, increased CGRP release at neutral pH in an ASIC3-dependent manner (please see new Supplementary Fig. 12d). However, at pH 6.5, their effect was not statistically significant. These results indicate that multiple LPC species may act as endogenous agonists of ASIC3 to activate sensory neurons, with variable effects.

Although other species of LPC may also contribute to the progression of psoriasis,

the available data suggest that 14:0 is the most changed LPC species in the lesion skin. Despite the lower abundance, the ratio of 14:0 to 18:1 LPC in the skin increased from 1:26 to 1:18 after IMQ treatment. The VIP (variable important in projection) score of LPC 14:0 (score 1.99) was larger than that of LPC 16:0 (score 1.31), 18:0 (score 1.44), and 18:1 (score 1.25) when considering the weights of these lipids as a variable, indicating that the enrichment in LPC 14:0 has a stronger ability to distinguish different samples.

Our data clearly suggest that 14:0 is a good representative LPC species to reveal the metabolic characteristics of psoriasis. With a molecular weight of 467.58, its concentration in the skin may reach μM levels, which is sufficient to trigger the opening of ASIC3 (please refer to new Fig. 7b and 7c). The *in vivo* data presented in Figure 7 also suggests that LPC14:0, when added alone, can worsen the pathological symptoms of psoriasis through ASIC3.

Reference

1. Marra S, *et al.* Non-acidic activation of pain-related Acid-Sensing Ion Channel 3 by lipids. *EMBO J* **35**, 414–428 (2016).
2. Jacquot F, *et al.* Lysophosphatidylcholine 16:0 mediates chronic joint pain associated to rheumatic diseases through acid-sensing ion channel 3. *Pain* **163**, 1999–2013 (2022).
8. *In Figure 7d, pH7.4 + LPC14:0 can trigger CGRP release via an ASIC3-dependent manner. Can LPC14:0 directly activate ASIC3 at neutral pH?*

Based on the reviewer's feedback, we have performed additional investigation on the facilitation effect of LPC14:0 on ASIC3. We found that although 10 μM LPC can only sensitize the acid-induced current of ASIC3 (please see Supplementary Fig. 12b,c), at 30 μM , LPC14:0 can directly activate whole-cell currents in ASIC3-

expressing cells at the physiological pH of 7.4 (please see new Fig. 7a and 7b). This result is consistent with the CGRP ELISA data in Figure 7c, in which pH 7.4 + 30 μ M LPC14:0 induced CGRP release from DRG neurons.

9. Is LPC16:0 detectable in skin samples?

Yes, LPC16:0 was detected in skin samples, but its level was not significantly higher in IMQ-treated mice than vehicle-treated ones. Furthermore, LPC16:0 did not induce CGRP release at neutral pH in *Asic3*^{+/+} DRG neurons (please see new Supplementary Fig. 12a,d).

Minor points:

1. In page 9, line 179. The sentence is confusing. The data were compared between conditional rescue of ASIC3 in Nav1.8-Cre mice and ASIC3 knockdown mice (but not WT mice).

Correct accordingly (please see Page 9, the 2nd paragraph).

2. In page 22, line 461. Is it exon 11? Or exon 1?

Correct accordingly (please see the Supplementary Fig. 5 and its legend).

3. A potential ethical issue is concerned. The Nav1.8-Cre mice were generated by Dr. Rohini Kuner's group. It is not appropriate to use the mice from Dr. Dr. Xu Zhang without acknowledgement of the original authors.

Correct accordingly (please see the Methods section).

4. The dose of Bicuculine is missing in the Methods (and Figure legends).

Done accordingly (please see Supplementary Fig. 7 and Methods section).

5. *In Supplementary Figure 10a, is the data only obtained from lesion skin? How about normal skin? How can the -0.55V be converted into pH values?*

Thanks for bringing up this point. We apologize for any confusion caused by the previous version of Supplementary Fig. 10a. We have now updated the figure and its legend to improve clarity. In the previous version, two samples, both from the IMQ-treated group, were shown, which made it difficult to view the change. We hope that the updated version helps clarify the method (please see the new Supplementary Fig. 10a). The theory for monitoring skin surface pH using Hemin-Fc/CNT microelectrodes is that the potential (E vs. Ag/AgCl) of Fe^{2+/3+} in hemoglobin shifts its peak value (redox peak) as the pH changes, and the changes in pH and peak potentials follow a strictly linear relationship. The pH value of normal skin tissue is around 7.5, and its peak potential reading by Hemin-Fc/CNT microelectrode is around - 0.6 V. By applying a linear equation, we can calculate the pH value of the lesion tissue based on the shift of the peak potential (from -0.6 V to -0.5 V). (*Angew. Chem. Int. Ed.*, 2017, 56, 10471-10475.)

Reference

1. Liu L, *et al.* An Electrochemical Biosensor with Dual Signal Outputs: Toward Simultaneous Quantification of pH and O₂ in the Brain upon Ischemia and in a Tumor during Cancer Starvation Therapy. *Angew Chem Int Ed Engl* **56**, 10471-10475 (2017).

Reviewer #2 (Remarks to the Author):

The authors have nicely addressed all the reviewer's comments with new experiments. The revised manuscript is suitable for publication at Nature Communications.

We thank the reviewer for the positive comments.

REVIEWERS' COMMENTS

Reviewer #1 (Remarks to the Author):

The authors have enthusiastically addressed all my comments and the manuscript has been largely improved. Two minor points are raised.

1. Page 23. Please make sure that in mouse *Asic3* gene, the ATG start codon is located at exon 11?
2. Please make sure you have the approval from Dr. Rohini Kuner to use the Nav1.8-Cre mice.

Responses to Reviewers' comments (Reviewers' comments in *italic*)

Reviewer #1:

The authors have enthusiastically addressed all my comments and the manuscript has been largely improved. Two minor points are raised.

We appreciate the positive feedback and previous suggestions from the reviewer.

1. *Page 23. Please make sure that in mouse *Asic3* gene, the ATG start codon is located at exon 11?*

Thanks for notifying us of the error. We have checked the information in the public database and corrected the mistake in 'exon 1'.

2. *Please make sure you have the approval from Dr. Rohini Kuner to use the *Nav1.8-Cre* mice.*

Thanks for your suggestion. We have contacted Professor Rohini Kuner and obtained her approval to use this strain of mice. With Professor Kuner's guidance, we have also confirmed the original publication literature to cite for this animal strain.